# Counterfactual Variable Control for Robust and Interpretable Question Answering

## Abstract

Deep learning based question answering (QA) models are neither robust nor interpretable in many cases. For example, a multiple-choice QA model, tested without any input of *question*, is surprisingly "capable" to predict most of correct *answers*. In this paper, we inspect such "shortcut capability" of the QA model using causal inference. We find the crux behind is the shortcut correlation (learned in the model), *e.g.*, simply word alignment between *passage* and *options*. To address the issue, we propose a novel approach called Counterfactual Variable Control (CVC) including both training and inference stages that explicitly mitigates any shortcut correlations and preserves only comprehensive reasoning to do robust QA. To enable CVC inference, we first leverage a multi-branch network architecture Cadene et al. (2019) based on which we disentangle shortcut correlations and comprehensive reasoning in the trained model in CVC training. Then, we introduce two variants of CVC inference approach to capture only the causal effect of comprehensive reasoning as the model prediction. To evaluate CVC, we conduct extensive experiments using three neural network backbones (BERT-base, BERT-large and RoBERTa-large) on both multi-choice and extractive QA benchmarks (MCTest, DREAM, RACE and SQuAD). Our results show that CVC can achieve consistently high robustness against various adversarial attacks in QA tasks, and its results are easy to interpret.

## 1 Introduction

Question answering (QA) is an important task in natural language processing that has been attracting much attention in recent years Seo et al. (2016); Chen et al. (2017); Yu et al. (2018); Kwiatkowski et al. (2019); Karpukhin et al. (2020); Yasunaga et al. (2021). Although tremendous progress has been made with QA models, especially with the help of pre-trained language models such as BERT Devlin et al. (2019) and RoBERTa Liu et al. (2019), top-performing models often lack interpretability Feng et al. (2018); Kaushik & Lipton (2018), nor are they robust to adversarial attacks Ribeiro et al. (2018); Szegedy et al. (2013); Wallace et al. (2019); Niu & Zhang (2021); Kiela et al. (2021). For example, adding one more question mark at the end of the input question, which is a simple adversarial attack, may decrease the performance of QA models Ribeiro et al. (2018). This vulnerability will raise security concerns when the model is deployed in real-world applications, *e.g.*, intelligent shopping assistants and web search engines. It is thus desirable to figure out why this happens and how to improve the robustness of QA models.

Existing methods for robust QA models mainly resort to robust training. One straightforward way is to generate adversarial examples for training Jia & Liang (2017); Ribeiro et al. (2018); Si et al. (2021). However, sometimes it is expensive and time-consuming to manually generate adversarial examples, and QA models are still not robust to unseen attacks. On the other hand, recent works focus on regularizing QA models via additional losses to prevent the model from learning the superficial correlation. For example, QAInformax Yeh & Chen (2019) maximizes the mutual information between the passage and the question to achieve regularization. However, so far, robust QA against adversarial samples has not been fully exploited.

In this paper, we carefully inspect both the training and the test processes for QA models. We find the aforementioned vulnerability is caused by the fact that the model tends to exploit the *shortcut correlations* in

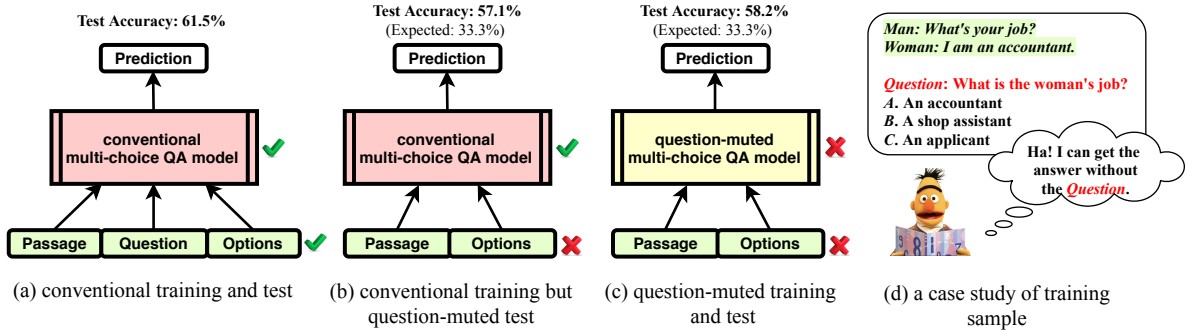

Figure 1: We observe multi-choice QA models are "capable" to answer a question without any *question* data in input (question-muted) during test (b), or during both training and test (c). We conduct these experiments using the BERT-base model Devlin et al. (2019) on the multi-choice QA benchmark DREAM Sun et al. (2019). (a) shows the normal case for reference. (d) show a training sample on DREAM.

the training data. To illustrate this, we show some example results of the BERT-Base MCQA model Devlin et al. (2019) in Figure 1. Supposedly, the model should predict the answer based on the passage, the question, and the options. Surprisingly, the absence of the *question* during only the test stage (Figure 1 (b)) or during both the training and the test stages (Figure 1 (c)) leads to a limited performance drop. Our hypothesis is that the BERT-Base MCQA model uses a huge amount of network parameters to learn the *shortcut correlation* between the *no-question* inputs (*i.e.*, *passage* and *options*) and the ground-truth *answer* in a brute-force manner. Figure 1(d) shows an example where this *shortcut* could be realized by simply aligning the words appearing in both the *passage* and *options*. Can we just conclude from this example that *questions* have little effect on *answers*? We must say no, as this violates our common sense about the causality in QA — *the question causes the answer*.

With the observation above in mind, we take a step further towards robust and interpretable QA systems by figuring out the causality in QA based on causal inference Pearl et al. (2009); Pearl & Mackenzie (2018). We begin by analyzing the causal relationships in QA, *i.e.*, associating any two variables based on the causal effect. Inspired by the recent success of causal inference in applications Qi et al. (2020); Tang et al. (2020); Niu et al. (2021), we represent the causal relationships in QA using the Structural Causal Model (SCM) Pearl et al. (2009). Figure 2(a) shows the SCM for MCQA as an example, where each node denotes a variable (*e.g.*, $Q$ for *question* and $A$ for *answer*) and the directed edge from one node to another represents their causal relation (*e.g.*, $Q \to A$ denotes *question causes answer*). Besides the input and output variables, we introduce an intermediate variable $R$ to reflect the expected comprehen-

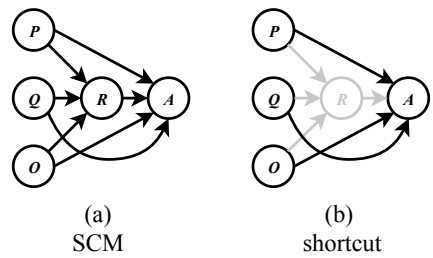

Figure 2: The SCM of MCQA. $P$ is for *passage*, $Q$ for *question*, $O$ for *options* and $A$ for *answer*. Particularly, $R$ denotes the comprehensive *reasoning*.

sive *reasoning* among all the inputs. SCM illustrates that not only comprehensive reasoning but also shortcut correlations have effects on the output answer. As highlighted in Figure 2(b), $P$ and $O$ can directly reach $A$, leading to a success rate 24% higher than the random guess shown in Figure 1(b). These shortcut correlations are "distractors" against our goal of robust QA, *i.e.*, the prediction should be caused by the comprehensive reasoning.

According to the above causality-based analysis, we expect the robust QA systems to conduct comprehensive reasoning and exclude the shortcut effects for unbiased inference. To alleviate the effects of shortcuts, we propose a novel approach called Counterfactual Variable Control (CVC) based on the causality theory. CVC in essence includes *counterfactual analysis* Pearl et al. (2009); Pearl & Mackenzie (2018); Pearl (2001) and *variable control*. The former allows us to evaluate the effect of an event by modifying it in a counterfactual scenario. The latter, motivated by controlling for variables, aims to explicitly separate the effects of different

variables. In this way, we can avoid any interference from controlled variables. To implement CVC in deep models, we realize the SCM as a multi-branch architecture Cadene et al. (2019); Clark et al. (2019) that is composed of a robust branch reflecting the comprehensive reasoning and several shortcut branches. We highlight that CVC training exactly follows the multi-branch training Cadene et al. (2019), while CVC is based on counterfactual analysis to capture the indirect effects of only the comprehensive reasoning. Furthermore, according to our causality-based analysis, we point out that existing ensemble-based debiasing methods Clark et al. (2019) can be regarded as special cases of CVC, which serves as a theoretical explanation from a perspective of causality. This is because these methods also involve multi-branch training, with robust and bias branches, and use only the robust branch during inference. To evaluate the robustness of CVC comprehensively, we propose four adversarial attacks for Multiple-Choice QA (MCQA) and one human-annotated adversarial set for Extractive QA (EQA). For example, we propose to add one sentence in passage as additional option to fool the MCQA model. Experiments are conducted on four QA benchmarks with different backbone networks, *e.g.*, BERT Devlin et al. (2019) and RoBERTa Liu et al. (2019). The experimental results validate the effectiveness and generalizability of our proposed CVC approach. As shown in the case studies, our CVC can not only achieve robust performance, but also conduct interpretable and reasonable inference processes due to the theoretical foundation of causal inference. Our main contributions are summarized as follows: (i) We analyze the vulnerability of QA model from a novel causal perspective and point out that the the robust prediction actually equals to the indirect effects of the input variables on answers. (ii) Based on the theory of causal inference, we propose counterfactual variable control (CVC) to measure the indirect effects, *i.e.*, mitigating the shortcut correlations while preserving the robust comprehensive reasoning in QA, and implement it in the deep models. Interestingly, our CVC method also cover the popular ensemble-based debiasing methods and provide interpretability for them. (iii) For a comprehensive evaluation, we propose several adversarial attacks for MCQA and EQA. The experimental results with different backbones on these adversarial sets for four QA benchmarks show the effectiveness and generality of CVC.

## 2  Related Work

**Question Answering.** Question answering (QA) is an important application in natural language understanding. QA aims to evaluate machines' reading comprehension abilities. Basically, the QA task requires machines to answer a question based on a given passage. Several QA settings have been proposed to inspect various aspects of language understanding. For example, CoQA Reddy et al. (2019) and DREAM Sun et al. (2019) are based on conversations, HotpotQA Yang et al. (2018) focuses on multi-hop reasoning, and RACE Lai et al. (2017) looks at the challenges with the multiple-choice QA setting. Meanwhile, the development of QA models has been rapid. Early QA models only relied on word embedding techniques such as Word2vec Mikolov et al. (2013) or GloVe Pennington et al. (2014). Inspired by the success of attention mechanism in machine translation Bahdanau et al. (2014), recent works further adopted this mechanism as core component, *e.g.*, BiDAF Seo et al. (2016) and Match-LSTM Wang & Jiang (2016). Nowadays, large-scale pre-training becomes indispensable in QA models, such as BERT Devlin et al. (2019) and RoBERTa Liu et al. (2019). However, robustness and interpretability of QA models are still challenging problems for practical applications Ribeiro et al. (2018); Wallace et al. (2019). In this paper, we focus on how to improve the robustness and interpretability of QA models. We take two types of QA datasets, multiple-choice question answering and span-extraction question answering, as study cases.

**Robustness in NLP.** Large-scale pre-trained language models have shown their strength in language understanding, however, it has been shown that many of them can be easily fooled by simple adversarial attacks, *e.g.*, using distractor sentences Zhang et al. (2020b). Recent works used generated adversarial examples to augment training data explicitly, such as to train more robust models against adversarial attacks Ribeiro et al. (2018); Liu et al. (2020a); Jia & Liang (2017); Wang & Bansal (2018). They achieved fairly good performance but they have their limitations. First, they need the prior knowledge of the specific adversarial attack, *i.e.*, "in what way to generate adversarial examples", which is often not available in real applications. Second, their model performance strongly relies on the quality of adversarial examples and is sensitive to training hyperparameters, *e.g.*, augmentation ratios. Alternative methods include using advanced regularizer Yeh & Chen (2019); Liu et al. (2020b); Ye et al. (2020), training loss Jia et al. (2019); Huang et al. (2019); Jiang et al. (2020), sample filtering Yaghoobzadeh et al. (2019); Le Bras et al. (2020) and

model ensembling Clark et al. (2019); Cadene et al. (2019); He et al. (2019); Utama et al. (2020a); Ghaddar et al. (2021). However, it remains unexplained why and how these methods are capable to achieve robust QA models.

In terms of model implementation, our CVC training process is close to ensemble-based methods. **Our key difference** is that we introduce a systematic and explainable causal formulation of QA that potentially opens principled directions to understanding the critical challenges of QA. Specifically, our first step was to formulate the QA task from a causal perspective using the Structural Causal Model (SCM) in order to understand the reasons behind the vulnerability of deep models. Building upon this analysis, we then proposed CVC to develop robust and interpretable QA models. CVC leverages the insights gained from the SCM analysis and offers an intuitive inference process for human interpretation. Our technical contribution lies in the new CVC methods (after the QA model is trained).

**Causal Inference in Deep Learning.** Causal inference Pearl et al. (2009); Pearl & Mackenzie (2018); Neuberg (2003) is based on the causal assumption made in each specific task, *e.g.*, QA task in this paper. It has been widely applied to epidemiology Rothman & Greenland (2005), computer science Van der Laan & Rose (2011), and social science Steel (2004). Recently, it has been incorporated in a variety of deep learning applications such as image classification Goyal et al. (2019), image parsing Zhang et al. (2020a), representation learning Wang et al. (2020), scene graph generation Tang et al. (2020), and vision-language tasks Qi et al. (2020); Chen et al. (2020); Niu et al. (2021); Abbasnejad et al. (2020). In NLP, counterfactual methods are also emerging recently in natural language inference Kaushik et al. (2020), semantic parsing Lawrence & Riezler (2018), story generation Qin et al. (2019), dialog systems Zhu et al. (2020), gender bias Vig et al. (2020); Shin et al. (2020), and sentiment bias Huang et al. (2020). In this paper, we take the first step towards improving the robustness of QA models based on causality.

## 3 Counterfactual Variable Control (CVC)

CVC aims to conduct unbiased inference by excluding the shortcut effects, *e.g.*, aligning words in passage and options. In this section, we continually use multi-choice question answering (MCQA) (and its SCM in Figure 2) as a case study of QA tasks and introduce our proposed Counterfactual Variable Control (CVC) on the level of SCM. In the next section, we turn to the implementation that how to model SCM and conduct the CVC using the deep model. Given a natural language paragraph as passage $p$, the models for MCQA are expected to answer the related question $q$ by selecting the correct answer $a$ from the candidate options $o$. In the following, we use uppercase letters to denote the variables (*e.g.*, $Q$ for *question*) and lowercase letters for the specific value of a variable (*e.g.*, $q$ for a specific question).

### 3.1 Normal Prediction and Counterfactual Prediction

We introduce counterfactual notations, *i.e.*, the imagined values of variables as if their ancestors had existed (*i.e.*, uncontrolled) in a counterfactual world Pearl et al. (2009); Tang et al. (2020); Pearl (2001); Roese (1997). For example, input variables ($P$, $Q$ and $O$) are set to the be available for $A$ while $R$ would attain the value when the input variables had been unavailable. We call this "counterfactual" as the variables cannot be simultaneously set as different statuses in the factual world.

**Normal Prediction (NP)** means that the model makes predictions when the variables are all controlled or uncontrolled. We use the function format $Y(X\!=\!x)$, abbreviated as $Y_x$, to represent the effect of $X\!=\!x$ on $Y$. We use this notation to formulate any path on the SCM, and further derive the prediction as:

$$A_{p,q,o,r} = A(P\!=\!p, Q\!=\!q, O\!=\!o, R\!=\!r), \tag{1}$$

where $r\!=\!R(P\!=\!p, Q\!=\!q, O\!=\!o)$ denotes the normal value of comprehensive reasoning, and $A_{p,q,o,r}$ denotes the inference logits of the model with realistic inputs values. If all the inputs are controlled (*e.g.*, muting their values as null), the value that $A$ would obtain can be represented as:

$$A_{p^*,q^*,o^*,r^*} = A(P\!=\!p^*, Q\!=\!q^*, O\!=\!o^*, R\!=\!r^*), \tag{2}$$

where $r^*\!=\!R(P\!=\!p^*, Q\!=\!q^*, O\!=\!o^*)$, and $A_{p^*,q^*,o^*,r^*}$ is the inference logits of the model with null values of input variables, which are denoted as $p^*$, $q^*$, and $o^*$.

**Counterfactual Prediction (CP)** means that the model predicts the answer when some variables are controlled, but the others are assigned counterfactual values obtained when these variables are uncontrolled. This is a key operation in the *counterfactual analysis* Pearl et al. (2009); Pearl & Mackenzie (2018); Pearl (2001). For example, we can control the input variables $P$, $Q$, and $O$ with their values to null (denoted as $p^*$, $q^*$, and $o^*$), and assign their child node $R$ with a counterfactual value $r = R(P=p, Q=q, O=o)$ obtained when the inputs $P$, $Q$, and $O$ were valid. Similarly, we can control $R$ as $r^*$ while assigning its parent nodes $P$, $Q$, and $O$ with counterfactual values $p$, $q$, and $o$.

To conduct CVC inference, we propose two variants of counterfactual control: (i) controlling only input variables; and (ii) controlling only the mediator variable. For (i), we formulate the value of $A$ as:

$$A_{p^*,q^*,o^*,r} = A(P=p^*, Q=q^*, O=o^*, R=r), \tag{3}$$

For (ii), we have:

$$A_{p,q,o,r^*} = A(P=p, Q=q, O=o, R=r^*). \tag{4}$$

### 3.2 CVC

As previously stated, our objective is to preserve only the robust prediction derived from comprehensive reasoning and exclude shortcut correlations. Consequently, the goal is to measure the effect of comprehensive reasoning $R$[1]. Motivated by the theory of causality Morgan & Winship (2015), CVC can be realized by comparing the fact and its counterpart, *i.e.*, estimating the difference between the normal prediction (NP) and the counterfactual prediction (CP). Intuitively, the importance of the effect of a variable on the resulting variable can be revealed by controlled experiments. If the difference between the experimental group and the control group is large, this variable may have a significant effect on the output. We utilize this conclusion from another view. If we have the prior that a variable is essential, we expect the difference to be large corresponding to this variable. In our case, we expect the difference corresponding to the comprehensive reasoning $R$ is large , *i.e.*, the model should rely on $R$ for inference. Following the definition in Section 3.1, the idea can be realized by controlling on either inputs (*e.g.*, $Q$) or mediator variables (*e.g.*, $R$). Therefore, CVC can be realized in two ways corresponding to the controlled variables: CVC on Input Variables (CVC-IV) and CVC on Mediator Variables (CVC-MV).

**CVC on Input Variables (CVC-IV)** is derived as:

$$\text{CVC-IV} = A_{p^*,q^*,o^*,r} - A_{p^*,q^*,o^*,r^*}, \tag{5}$$

where in $A_{p^*,q^*,o^*,r}$ the input variables are *controlled* to be null (*e.g.*, $p^*$) while the mediator variable is set as its counterfactual value, which is obtained by imaging a counterfactual world where the inputs had not been controlled (*i.e.*, $r = R(p, q, o)$). In this case, the first term is CP and the second term is NP. We measure the effect of $R$ by comparing the two scenarios where the states of $R$ are different.

**CVC on Mediator Variable (CVC-MV)** is derived as:

$$\text{CVC-MV} = A_{p,q,o,r} - A_{p,q,o,r^*}, \tag{6}$$

where in $A_{p,q,o,r^*}$ the input variables are set as their observed values (*e.g.*, $p$) while the mediator variable is *controlled* by imagining a counterfactual world where all inputs had been set to null (*i.e.*, $r^* = R(p^*, q^*, o^*)$). In this case, the first term is NP and the second term is CP. We measure the effect of $R$ by comparing the two scenarios where the states of $R$ are different.

Note that both CVC-IV and CVC-MV aim to capture the causal effect of comprehensive *reasoning* in QA. The main difference lies in *on which variables to apply the control*. The surgery is on the input variables in CVC-IV and the mediator variable in CVC-MV. The former aims to remove all the shortcut correlations, while the latter preserves only the effect of comprehensive *reasoning* on answer after the subtraction.

---

[1]Noted that $R$ variable is a virtual variable compared to other variables and denotes the robust and comprehensive reasoning.

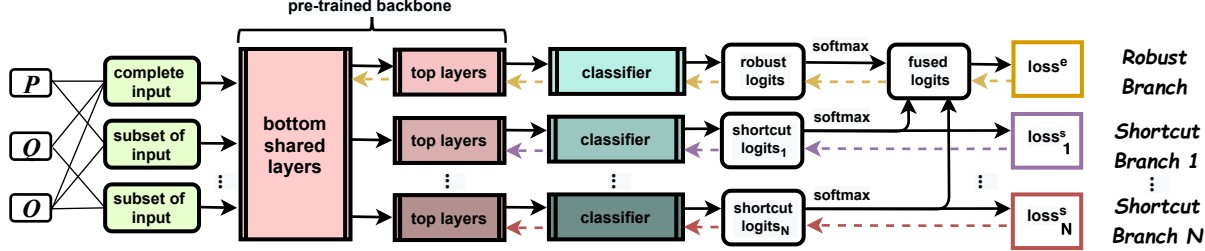

Figure 3: Multi-task training framework in our CVC using MCQA as example. It shows a robust branch and two shortcut branches ($N = 2$). The complete input (*e.g.*, $\mathcal{X} = \{P, Q, O\}$ for MCQA) are fed to the robust branch, while a subset to the shortcut branch (*e.g.*, $\mathcal{X}_1 = \{P, O\}$ to the first shortcut branch). Solid arrows indicate feedforward, and dashed arrows for backpropagation.

## 4 The Implementation of CVC

In this section, we introduce how to implement CVC using deep neural networks, including multi-task training and counterfactual inference strategies. Multi-task training explicitly decouples the robust path and shortcut paths by multi-branch architecture, while counterfactual inference conducts unbiased inference based on CVC-IV or CVC-MV in Section 3. Building on Section 3, we take MCQA and its corresponding SCM in Figure 2 as the example in this section.

### 4.1 Multi-task Training

As illustrated in Figure 3, our overall framework implements the SCM in Figure 2(a) as multiple neural network branches. The main branch takes all the input variables (*i.e.*, complete input) to learn the causal effect corresponding to the robust path of SCM (*i.e.*, $P, Q, O \rightarrow R \rightarrow A$), which we call comprehensive reasoning branch (or robust branch). The other branches, we call shortcut branches, take a subset of inputs (*i.e.*, part of the variables are muted) to explicitly learn the shortcut correlations corresponding to the shortcut paths of SCM (*e.g.*, $P, O \rightarrow A$ as $Q$ is muted). We deploy each branch as the standard QA model with pre-trained backbone Devlin et al. (2019) where the pre-trained backbone consists of bottom shared layers and top layers. The model is trained via multi-task training, *i.e.*, each branch is optimized using an individual objective. Only the robust branch gradients are propagated to update the bottom shared layers in the backbone. In the following, we introduce the details of how we implement each path in SCM (Figure 2).

$P, Q, O \rightarrow R \rightarrow A$. This path is implemented by the robust branch $F^r$, which takes the complete input $\mathcal{X} = \{P, Q, O\}$, *e.g.*, the realistic values of *question*, *passage* and *options* in MCQA. The network body, with parameters denoted as $\theta^r$, consists of a pre-trained backbone (*e.g.*, BERT) and a classifier (*e.g.*, one FC layer). Since this branch learns the causal effect from $R$ to $A$, we denote its output logits as:

$$A^r = F^r(\mathcal{X}; \theta^r). \tag{7}$$

We explain that R can be regarded as the hidden state of the top layers before the classifier on the robust branch. This illustrates the features of the text after being processed by the deep model's comprehensive reasoning ability.

$P \rightarrow A$, $Q \rightarrow A$ and $O \rightarrow A$. These paths are implemented by shortcut branches $F^s_n$ ($n = 1, 2, \cdots, N$), which learn the shortcut correlations between incomplete (controlled) input and the ground truth answer. Each shortcut branch takes a subset of variables $\mathcal{X}_n \subset \mathcal{X}$ as input (*e.g.*, $\mathcal{X}_1 = \{P, O\}$) and sets the other variables (*e.g.*, $Q$) as null. The shortcut branch has the same architecture with the robust branch but different parameters $\theta^s_n$. We denote the outputs as:

$$A^s_n = F^s_n(\mathcal{X}_n; \theta^s_n), \quad n = 1, 2, \cdots, N. \tag{8}$$

**Fusion** combines all causal effects from any variables $X$ directed linked to $A$, *e.g.*, $R \rightarrow A$ in robust branch and $O \rightarrow A$ in shortcut branch. Another functionality of the fusion is to facilitate the training of multi-branches.

We fuse the logits from the robust branch and shortcut branches as:

$$A_i^e = \sum_n \hat{\mathtt{p}}_i^r \cdot \hat{\mathtt{p}}_{n,i}^s, \tag{9}$$

where $\hat{\mathtt{p}}_i^r = \mathrm{softmax}(A_i^r)$, $\hat{\mathtt{p}}_{n,i}^s = \mathrm{softmax}(A_{n,i}^s)$ and $i$ is the $i$-th dimension of the prediction. Here we use probabilities instead of logits because probabilities are non-negative and can work as normalization.

**Objective** adopts the standard cross-entropy loss to optimize all the branches. For the $n$-th shortcut branch, we directly minimize the cross-entropy loss over its logits $A_n^s$:

$$\mathcal{L}_n^s = -\sum_i \mathtt{p}_i \log \mathrm{softmax}(A_{n,i}^s), \tag{10}$$

where $i$ denotes the $i$-th dimension of the prediction, and the one-hot vector $\mathtt{p}$ denotes the encoding of ground truth answer.

For the robust branch, directly optimizing over the robust prediction $A^r$ cannot avoid the model to learn the correlations as in the conventional QA models, and cannot guarantee the model to learn the pure comprehensive reasoning. We tackle this problem by fusing robust logits $A^r$ with shortcut logits $A_n^s$. In this way, we can force $A^r$ to only preserve the prediction that can never be achieved by shortcuts, *i.e.*, the comprehensive reasoning prediction with the complete input variables as input. We then optimize the cross-entropy loss over the adjustment $A^e$, *i.e.*, overall effects on $A$, for robust branch as:

$$\mathcal{L}^e = -\sum_i \mathtt{p}_i \log \mathrm{softmax}(A_i^e). \tag{11}$$

In pre-experiments, we empirically found that the robust branch may focus on only hard samples and ignore easy samples by fusing the branches at the level of predictions. When outputs of shortcut branches are correct with high confidence, logits-level fusion may lead to a very small value in Eq. 11. We further propose two variants of losses to tackle this issue:

$$
\begin{aligned}
\mathcal{L}^{e1} &= -\sum_n \frac{1}{n} \sum_i \mathtt{p}_i \log \mathrm{softmax}(\hat{\mathtt{p}}_i^r \cdot \hat{\mathtt{p}}_{n,i}^s), \\
\mathcal{L}^{e2} &= -\sum_n w_n \sum_i \mathtt{p}_i \log \mathrm{softmax}(\hat{\mathtt{p}}_i^r \cdot \hat{\mathtt{p}}_{n,i}^s),
\end{aligned}
\tag{12}
$$

where $w_n = \mathrm{softmax}(\mathcal{L}_n^s) = \frac{\exp(\mathcal{L}_n^s)}{\sum_{m=1}^{m=n} \exp(\mathcal{L}_m^s)}$ is a weight to explicitly enhance the effect of the $n$-th shortcut branch on the robust branch. We formulate the overall loss used in multi-task training as:

$$\mathcal{L}^{all} = \mathcal{L}^e + \sum_n \mathcal{L}_n^s, \tag{13}$$

where $\mathcal{L}^e$ can be replaced with $\mathcal{L}^{e1}$ or $\mathcal{L}^{e2}$. Ablation studies empirically show that $\mathcal{L}^{e2}$ achieves better performances. Noted that the optimization of robust branch will not affect the parameters on the shortcut branches.

### 4.2 Counterfactual Inference

Different from conventional inference that is based on the posterior probability Devlin et al. (2019), we propose to use counterfactual inference based on causal effects Pearl & Mackenzie (2018); Pearl (2001). In this section, we introduce how to conduct CVC-IV and CVC-MV inferences given the robust branch and shortcut branches.

Following the notation formats of normal prediction (NP) and counterfactual prediction (CP) in Eq. 3 and Eq. 4 along with the notation of output for each branch in Eq. 7 and Eq. 8, we can (i) denote the prediction of the $n$-th shortcut branch as $a_n^s = F_n^s(p, o; \theta_n^s)$ and its value with null input as $a_n^{s*} = F_n^s(p^*, o^*; \theta_n^s)$;

and (ii) denote the prediction of the robust branch as $a^r = F^r(p, q, o; \theta^r)$ and its value with null input as $a^{r*} = F^r(p^*, q^*, o^*; \theta^r)$. In this case, we denote $A_{a_1^{s*}, \cdots, a_N^{s*}, a^{r*}}$ as $A_{p^*, q^*, o^*, r^*}$.

For the CVC-IV inference in Eq. 5, we obtain NP as $A_{p^*, q^*, o^*, r^*}$ and CP as $A_{p^*, q^*, o^*, r}$. Combining Eq. 5 and 9, we can derive the **CVC-IV inference result** as:

$$
\begin{aligned}
\text{CVC-IV} &= A_{p^*, q^*, o^*, r} - A_{p^*, q^*, o^*, r^*} \\
&= A_{a_1^{s*}, \cdots, a_N^{s*}, a^r} - A_{a_1^{s*}, \cdots, a_N^{s*}, a^{r*}} \\
&= \sum_n \hat{\mathrm{p}}^r \cdot c_n^s - \sum_n c_n^r \cdot c_n^s,
\end{aligned}
\tag{14}
$$

where each element in $c_n^r$ or $c_n^s$ is the same constant in $[0, 1]$. Each element in $c_n^r$ or $c_n^s$ are the same constant derived from Eq. 9. Specifically, $c_n^r$ denotes the value of $\hat{\mathrm{p}}^r$ when the corresponding robust branch is fed null input, and thus the same constant is used to denote the logits in $c_n^r$. Similarly, $c_n^s$ corresponds to the value of $\hat{\mathrm{p}}_n^s$ when the corresponding shortcut branch is fed null input. We highlight that CVC-IV inference corresponds to computing Natural Indirect Effect (NIE) in causal inference Pearl & Mackenzie (2018); Pearl (2001). It is equivalent to the normal inference on the robust model, similar to existing works such as Learned-Mixin Clark et al. (2019) and RUBi Cadene et al. (2019). Differently, CVC-IV is totally derived from the *systematical* causal analysis in QA and is thus more *explainable* than Learned-Mixin which is heuristic.

For the CVC-MV inference in Eq. 6, we denote NP as $A_{p, q, o, r}$, and CP as $A_{p, q, o, r^*}$. Combining Eq. 6 and 9, we can derive the **CVC-MV inference result** as:

$$
\begin{aligned}
\text{CVC-MV} &= A_{p, q, o, r} - A_{p, q, o, r^*} \\
&= A_{a_1^s, \ldots, a_N^s, a^r} - A_{a_1^s, \ldots, a_N^s, a^{r*}} \\
&= \sum_n \hat{\mathrm{p}}^r \cdot \hat{\mathrm{p}}_n^s - \sum_n c_n^r \cdot \hat{\mathrm{p}}_n^s,
\end{aligned}
\tag{15}
$$

which is an indirect way of making inference using only the robust branch. This result corresponds to computing Controlled Indirect Effect (CIE) in causal inference Pearl & Mackenzie (2018); Pearl (2001).

Since the optimal value for $c_n^r$ varies across each sample, we train a $c$-adaptor $F_n^c$ with a two-layer MLP to adaptively estimate $c_n^r$. This can be formulated as:

$$
c_n^r = F_n^c(\hat{\mathrm{p}}^r, \hat{\mathrm{p}}_n^s, Distance; \theta_n^c),
\tag{16}
$$

where $F_n^c(x_1, x_2, x_3; \theta_n^c) = \mathbf{W}_n^2 \tanh(\mathbf{W}_n^1[x_1; x_2; x_3])$, $[;]$ is the concatenation operation, and $\theta_n^c = \{\mathbf{W}_n^1, \mathbf{W}_n^2\}$ are learnable parameters. We implement $Distance$ as the Jensen-Shannon divergence Lin (1991) $\mathbf{JS}[\hat{\mathrm{p}}^r || \hat{\mathrm{p}}_n^s]$ between $\hat{\mathrm{p}}^r$ and $\hat{\mathrm{p}}_n^s$. Specifically, we train a $c$-adaptor after the multi-task training in Figure 3 by fixing the other parameters. The training objective is the same as the downstream task, *e.g.*, computing the cross-entropy loss with the logits of CVC-MV (Eq. 6) and ground-truth label:

$$
\mathcal{L}^{c-adapter} = - \sum_i \mathrm{p}_i \log \mathrm{softmax}(\sum_n \hat{\mathrm{p}}^r \cdot \hat{\mathrm{p}}_n^s - \sum_n c_n^r \cdot \hat{\mathrm{p}}_n^s).
\tag{17}
$$

### 4.3 Summary

Our approach consists of two stages: multi-task training (Section 4.1) and counterfactual inference (Section 4.2) summarized in Algorithm 1 (in Appendix). Multi-task training aims to train a robust branch $F^r$ and $N$ shortcut branches $\{F_n^s\}_{n=1}^N$. Counterfactual inference performs the robust and interpretable reasoning for QA. We highlight that CVC training follows the supervised training on multi-task networks Cadene et al. (2019); Clark et al. (2019). CVC differs from the inference process described in Section 4.B. Normal prediction will use the overall trained model directly for the inference. Some debiasing method Cadene et al. (2019); Clark et al. (2019) adopt the prediction only from the robust branch and discard the bias branch. The inference process of CVC is derived from the causal analysis of the shortcut problems in QA model. Furthermore, the inference process in Cadene et al. (2019); Clark et al. (2019) (directly using the robust branch) can be regarded as a special case of CVC-IV while lacks interpretability.

## 5 Experiments

### 5.1 Experimental Settings

We evaluate the robustness of CVC for both MCQA and EQA, using a variety of adversarial attacks Zhang et al. (2020b). Below we introduce the base datasets followed by the adversarial sets for each base datasets. We conduct multi-task training on the training split of base datasets and conduct inference on original development/test splits of base datasets and adversarial sets.

#### 5.1.1 Base Datasets

MCQA aims to select the correct answer from several input options given a passage and a question. We conduct experiments on the following benchmark datasets: **MCTest** Richardson et al. (2013) is generated from fictional stories and aims at open-domain machine comprehension. The questions are limited to the level that young children can understand. MCTest consists of two subsets, MC500 and MC160. We use the combination of them in our experiments. **DREAM** Sun et al. (2019) is a dialogue-based dataset designed by experts to evaluate the comprehensive ability of foreign learners. In addition to simply matching questions, DREAM also contains more challenging questions that requires commonsense reasoning. **RACE** Lai et al. (2017) is a dataset of English exam from middle and high school reading comprehension. RACE covers a variety of topics and the proportion of questions that requires reasoning is much larger than other reading comprehension datasets.

Compared to MCQA, options are not provided on the EQA task. EQA locates the answer span in a passage given a question. We use the SQuAD dataset for EQA. **SQuAD** Rajpurkar et al. (2016) is adopted as the benchmark for EQA where passages are from a set of Wikipedia articles. SQuAD requires several types of reasoning like lexical variation, syntactic variation, etc.

#### 5.1.2 Adversarial Sets

**Adversarial Attacks on MCQA.** To further evaluate the robustness of QA models, we propose four kinds of grammatical adversarial attacks to generate adversarial examples. `Add1Truth2Opt` and `Add2Truth2Opt` (`Adv1` and `Adv2`): We replace one (or two) of the wrong options with another one (or two) answers that are correct in other samples with the same passage. `Add1Pas2Opt` (`Adv3`): We replace one of the wrong options with a random distracting sentence extracted from the passage. This distractor does not contain any word that appears in the ground truth option. `Add1Ent2Pas` (`Adv4`): We first choose one of the wrong options with at least one entity, *e.g.*, person name and time, and then replace each entity with another entity of the same type. Then, we add this modified sentence to the end of the passage.

**Adversarial Attacks on EQA.** For the EQA task, we utilize three kinds of grammatical adversarial attacks. `AddSent` (`Adv1`), `AddOneSent` (`Adv2`) and `AddVerb` (`Adv3`). `AddSent` and `AddOneSent` released by Jia & Liang (2017) add distracting sentences to the passage. The generating process is: firstly perturb the question (*e.g.*, asking another entity) and create a fake answer, then convert the perturbed question into a distracting sentence. The final distracting sentences were filtered by crowdworkers. `AddSent` is similar to `AddOneSent` but much harder than `AddOneSent`. These two settings can be used to measure the model robustness against *entity* or *noun* attacks. `AddVerb` was inspired by above two sets which aims to evaluate the model robustness against *verb* attacks instead of *noun*. we hire expert linguists to annotate the `AddVerb` following Jia & Liang (2017). Examples are as follows. For the question *"What city did Tesla move to in 1880?"*, `AddSent` sample could be *"Tadakatsu moved to the city of Chicago in 1881."*, and `AddVerb` sample could be *"Tesla left the city of Chicago in 1880."*

### 5.2 Implementation Details

We illustrate the general implementation here and more details for MCQA-specific and EQA-specific are placed in the appendix. We deploy the pre-trained BERT and RoBERTa backbones provided by HuggingFace Wolf et al. (2019). The learning rates are fixed to 3e-5, 2e-5 and 1e-5 for BERT-base, BERT-large, and RoBERTa-large respectively. A linear warm-up strategy for learning rates is used with the first 10% steps in the whole

multi-branch training stage. The batch size is selected amongst $\{16, 24, 32\}$ for the three backbones. The number of bottom shared layers is fixed to 5/6 of the total number of layers in the backbone language model for parameter-efficiency, *e.g.*, sharing 10 layers in bottom shared layers when the BERT-base (12 layers) is adopted as the backbone. The overall experiments are conducted on two pieces of Tesla V100 or two pieces of RTX 2080Ti (depending on the usage of memory). Gradient accumulation and half precision are used to relieve the issue of memory usage. Following Clark et al. (2019); Grand & Belinkov (2019); Ramakrishnan et al. (2018), we perform model selection for CVC-IV (*i.e.*, choosing the hyperparameters of training epochs) based on the model performance in the development/test sets on the used dataset. We report the average performance with four random seeds.

| Set | Method | BERT-base | | | | | | BERT-large | | | | | | RoBERTa-large | | | | | |
|---|---|---|---|---|---|---|---|---|---|---|---|---|---|---|---|---|---|---|---|
| | | Test | Adv1 | Adv2 | Adv3 | Adv4 | A.G. | Test | Adv1 | Adv2 | Adv3 | Adv4 | A.G. | Test | Adv1 | Adv2 | Adv3 | Adv4 | A.G. |
| MCTest | CT | 68.9 | 63.9 | 59.4 | 20.2 | 54.8 | - | 72.3 | 70.0 | 66.8 | 35.5 | 57.6 | - | 88.9 | 88.2 | 86.6 | 72.6 | 84.2 | - |
| | CVC-MV | 68.1 | 69.1 | 65.6 | 26.8 | 61.0 | **+6.1** | 73.2 | 74.3 | 73.5 | 38.4 | 68.4 | **+6.2** | 88.5 | 89.3 | 89.6 | 82.4 | 83.4 | **+3.3** |
| | CVC-IV | 69.4 | 70.0 | 65.4 | 28.7 | 59.9 | **+6.4** | 74.4 | 75.5 | 75.1 | 40.4 | 69.5 | **+7.6** | 87.4 | 88.1 | 88.2 | 82.6 | 84.2 | +2.9 |
| DREAM | CT | 61.5 | 47.5 | 39.2 | 20.9 | 41.8 | - | 65.9 | 50.6 | 43.0 | 25.6 | 48.2 | - | 84.1 | 78.2 | 76.3 | 57.1 | 71.8 | - |
| | CVC-MV | 60.1 | 49.6 | 39.9 | 23.7 | 45.6 | **+2.3** | 64.0 | 51.9 | 46.5 | 26.3 | 51.3 | **+2.2** | 82.8 | 77.9 | 80.2 | 66.6 | 71.4 | +3.2 |
| | CVC-IV | 60.0 | 49.2 | 40.7 | 25.0 | 47.1 | **+3.1** | 64.5 | 52.0 | 46.2 | 26.6 | 51.1 | +2.1 | 81.7 | 78.3 | 79.7 | 66.7 | 72.3 | **+3.4** |
| RACE | CT | 64.7 | 56.0 | 50.1 | 36.6 | 58.3 | - | 67.9 | 61.9 | 57.9 | 51.0 | 61.7 | - | 78.4 | 72.4 | 67.9 | 65.9 | 72.1 | - |
| | CVC-MV | 64.4 | 56.7 | 51.7 | 39.1 | 59.2 | **+1.4** | 68.6 | 63.1 | 58.0 | 52.4 | 65.3 | +1.6 | 77.9 | 75.1 | 72.0 | 68.1 | 72.6 | +2.4 |
| | CVC-IV | 64.1 | 57.0 | 52.2 | 38.8 | 58.6 | +1.4 | 68.4 | 62.9 | 58.7 | 51.8 | 65.6 | **+1.6** | 77.4 | 75.7 | 73.8 | 69.1 | 72.1 | **+3.1** |

Table 1: Accuracies (%) on three MCQA datasets. Models are trained on original training data. BERT-base, BERT-large and RoBERTa-large are backbones. "A.G." denotes the average improvement over the conventional training (CT) Devlin et al. (2019) for `Adv*` sets.

| | BERT-base | | | | | BERT-large | | | | | RoBERTa-large | | | | |
|---|---|---|---|---|---|---|---|---|---|---|---|---|---|---|---|
| Method | Dev | Adv1 | Adv2 | Adv3 | A.G. | Dev | Adv1 | Adv2 | Adv3 | A.G. | Dev | Adv1 | Adv2 | Adv3 | A.G. |
| CT | 88.4 | 49.9 | 59.7 | 44.6 | - | 90.6 | 60.2 | 70.0 | 50.0 | - | 93.5 | 77.0 | 82.8 | 61.3 | - |
| QAInformax | 88.6 | 54.5 | 64.9 | - | +4.9 | - | - | - | - | - | - | - | - | - | - |
| MASS | 80.2 | 53.4 | 63.5 | - | +3.7 | - | - | - | - | - | - | - | - | - | - |
| ECS-P | 88.4 | 55.2 | 65.5 | - | +5.8 | - | - | - | - | - | - | - | - | - | - |
| ECS-Q | 87.4 | 54.4 | 65.0 | - | +4.9 | - | - | - | - | - | - | - | - | - | - |
| CVC-MV | 87.2 | 55.7 | 65.3 | 51.3 | +6.0 | 90.2 | 62.6 | 72.4 | 52.5 | +2.4 | 92.6 | 79.4 | 84.1 | 63.2 | +1.9 |
| CVC-IV | 86.6 | 56.3 | 66.2 | 51.5 | **+6.6** | 89.4 | 62.6 | 71.8 | 54.1 | **+2.8** | 92.2 | 79.6 | 85.0 | 64.1 | **+2.5** |

Table 2: EQA F1-measure (%) on the SQuAD `Dev` set (`Test` set is not public) and adversarial sets. Models are trained on original training data. BERT-base, BERT-large and RoBERTa-large are backbones. "-": not applicable from original paper. "A.G.": our average improvement over the conventional training (CT) Devlin et al. (2019) for `Adv*`. Results of ECS and MASS are from Xu et al. (2022) and Majumder et al. (2021).

## 5.3 Results and Analyses

**Comparison with Baselines and State-of-the-Arts.** Table 1 and Table 2 show the overall results for MCQA and EQA, respectively. Note that the adversarial sets `Adv` are used to evaluate the robustness of QA models. We report the average gain on `Adv`, denoted as A.G., to compare CVC with the conventional training methods (CT). From Table 1, we can see that both CVC-MV and CVC-IV can surpass the baseline method Devlin et al. (2019) for defending against adversarial attacks, *e.g.*, by average increase of 7.6% with BERT-large and 3.3% with RoBERTa-large on MCTest. It is worth highlighting the example that CVC-IV on

| Set | Method | Test | Adv1 | Adv2 | Adv3 | Adv4 | A.G. |
|---|---|---|---|---|---|---|---|
| MCTest | CT Devlin et al. (2019) | 68.9 | 63.9 | 59.4 | 20.2 | 54.8 | - |
| | DRiFt He et al. (2019) | 69.6 | 66.0 | 61.9 | 23.0 | 54.8 | +1.9 |
| | Bias Product Clark et al. (2019) | 71.0 | 66.7 | 63.6 | 22.8 | 65.5 | +5.1 |
| | Learned-Mixin Clark et al. (2019) | 70.5 | 66.2 | 60.4 | 20.2 | 58.8 | +1.8 |
| | Unknown Bias Utama et al. (2020b) | 68.1 | 64.5 | 62.7 | 20.7 | 59.1 | +2.2 |
| | Self-Debiasing Ghaddar et al. (2021) | 69.2 | 64.9 | 61.5 | 22.2 | 57.2 | +1.9 |
| | CVC-MV | 68.1 | 69.1 | 65.6 | 26.8 | 61.0 | +6.1 |
| | CVC-IV | 69.4 | 70.0 | 65.4 | 28.7 | 59.9 | **+6.4** |
| DREAM | CT Devlin et al. (2019) | 61.5 | 47.5 | 39.2 | 20.9 | 41.8 | - |
| | DRiFt He et al. (2019) | 60.1 | 48.5 | 42.2 | 23.9 | 44.7 | +2.5 |
| | Bias Product Clark et al. (2019) | 58.6 | 47.5 | 38.8 | 22.6 | 40.2 | -0.1 |
| | Learned-Mixin Clark et al. (2019) | 60.9 | 49.2 | 41.7 | 20.0 | 42.3 | +1.0 |
| | Unknown Bias Utama et al. (2020b) | 59.3 | 48.7 | 40.3 | 24.5 | 43.2 | +1.8 |
| | Self-Debiasing Ghaddar et al. (2021) | 61.2 | 47.3 | 39.7 | 22.9 | 44.1 | +1.2 |
| | CVC-MV | 60.1 | 49.6 | 39.9 | 23.7 | 45.6 | +2.3 |
| | CVC-IV | 60.0 | 49.2 | 40.7 | 25.0 | 47.1 | **+3.1** |
| RACE | CT Devlin et al. (2019) | 64.7 | 56.0 | 50.1 | 36.6 | 58.3 | - |
| | DRiFt He et al. (2019) | 62.0 | 56.1 | 53.3 | 39.3 | 58.3 | **+1.7** |
| | Bias Product Clark et al. (2019) | 62.3 | 56.7 | 53.3 | 37.0 | 56.8 | +1.0 |
| | Learned-Mixin Clark et al. (2019) | 64.3 | 56.5 | 51.9 | 38.0 | 60.1 | +1.4 |
| | Unknown Bias Utama et al. (2020b) | 63.3 | 57.1 | 52.5 | 37.5 | 58.1 | +1.1 |
| | Self-Debiasing Ghaddar et al. (2021) | 63.5 | 56.6 | 52.7 | 38.2 | 58.9 | +1.4 |
| | CVC-MV | 64.4 | 56.7 | 51.7 | 39.1 | 59.2 | +1.4 |
| | CVC-IV | 64.1 | 57.0 | 52.2 | 38.8 | 58.6 | +1.4 |

Table 3: Comparison of ours and related ensembling methods on MCQA with BERT-base. We implement these methods by replacing Eq. 9 with their adjustment functions for known bias methods. "A.G.": our average improvement over the conventional training method (CT) Devlin et al. (2019) for `Adv*`.

| Method | Dev | Adv1 | Adv2 | Adv3 | A.G. |
|---|---|---|---|---|---|
| CT Devlin et al. (2019) | 88.4 | 49.9 | 59.7 | 44.6 | - |
| DRiFt He et al. (2019) | 85.7 | 53.7 | 65.7 | 48.5 | +4.5 |
| Bias Product Clark et al. (2019) | 87.8 | 53.6 | 65.7 | 47.3 | +4.1 |
| Learned-Mixin Clark et al. (2019) | 87.2 | 53.1 | 63.9 | 45.5 | +2.1 |
| Unknown Bias Utama et al. (2020b) | 88.2 | 50.3 | 62.7 | 47.8 | +2.2 |
| Self-Debiasing  Ghaddar et al. (2021) | 89.3 | 52.7 | 64.1 | 46.0 | +2.9 |
| CVC-MV | 87.2 | 55.7 | 65.3 | 51.3 | +6.0 |
| CVC-IV | 86.6 | 56.3 | 66.2 | 51.5 | **+6.6** |

Table 4: Comparison of ours and related ensembling methods on EQA with BERT-base. We implement DRiFt by directly changing our adjustment function (Eq. 9) to its. For Bias Product and Learned-Mixin, we first use the corresponding adjustment functions in Clark et al. (2019), then we use the TF-IDF released by original paper as the shortcut branch in our implementation for known bias methods. "A.G.": our average improvement over the conventional training method (CT) Devlin et al. (2019) for `Adv*`.

BERT-base gains 8.5% on the most challenging `Adv3` set of MCTest. Besides, our methods are applicable to different backbones like BERT and RoBERTa-large. The results on EQA in Table 2 show similar observation. These results empirically demonstrate that our CVC strategy is general and model-agnostic.

Compared to state-of-the-art method, our CVC is more robust to adversarial attacks. As shown in Table 2, CVC outperforms the state-of-the-art QAInformax Yeh & Chen (2019) by an average of 1.7% F1-measure with the same BERT-base backbone. As shown in Table 3 and Table 4, we compare CVC with ensemble-based

methods. These methods mainly differ in the way of ensembling during training and the design of the bias model. For example, Clark et al. (2019) adopts a TF-IDF model as the bias model on EQA while Utama et al. (2020b) employs an early-stopping model as the bias model. The results show that CVC outperforms these methods on MCTest and DREAM datasets. Besides, all the approach achieve less improvement on RACE compared to other two datasets. The possible reason is that RACE is designed for reading comprehension that highlights comprehensive reasoning. Thus, the training data is more debiased. Note that our counterfactual analysis can regard these ensemble based methods as implementation of our CVC-IV. Also, we notice that CVC-MV often performs worse than CVC-IV on `Adv` sets but better on in-domain `Test` (or `Dev`) sets. The possible reason is that the important hyperparameter of CVC-MV $c_n^r$ is learned from in-domain data. We will show that augmenting in-domain data with `Adv` examples greatly improves the performance of CVC-MV in Table 9.

**Ablation Study.** Table 5 shows the EQA results in 10 ablative settings to evaluate the importance of shortcut branches, loss functions, and inference strategies: (1) removing the first shortcut branch ($E$ muted) from the multi-task training; (2) removing the second shortcut branch ($V$ muted) from the multi-task training; (3) using $\mathcal{L}^e$ to replace $\mathcal{L}^{e2}$; (4) using $\mathcal{L}^{e1}$ to replace $\mathcal{L}^{e2}$; (5) setting $c_n^r$ to the same constant (tuned in $\{0.2, 0.4, 0.6, 0.8, 1\}$) for all input samples; (6) setting $c_n^r = \mathbf{JS}[\hat{p}^r||\hat{p}_n^s]$ where $\mathbf{JS}$ denotes Jensen–Shannon divergence; (7) setting $c_n^r$ as the euclidean distance between $\hat{p}^r$ and $\hat{p}_n^s$; (8) removing the *distance* item in Eq. 16 and (9) removing $\hat{p}^r$ and $\hat{p}_n^s$ in Eq. 16. Compared to the ablative results, we can see that our full approach achieves the overall top performance on EQA. There is one exception. A higher score on `Adv1` is achieved (57.7 vs. 55.7) if we do not use the second shortcut branch ($V$ muted), *i.e.*, the second ablative. However, this setting achieves much lower performance on `Adv3` (42.1 vs. 51.3). This

| Ablative Setting | Dev | Adv1 | Adv2 | Adv3 |
|---|---|---|---|---|
| (1) w/o first Shct.br. | 85.5 | 52.6 | 62.5 | 50.8 |
| (2) w/o second Shct.br. | 86.1 | 57.7 | 66.1 | 42.1 |
| (3) use $\mathcal{L}^e$ | 72.4 | 45.9 | 54.9 | 42.6 |
| (4) use $\mathcal{L}^{e1}$ | 86.5 | 53.5 | 63.2 | 46.7 |
| CVC-IV (ours) | 86.6 | 56.3 | 66.2 | 51.5 |
| (5) same $c_n^r$ | 85.7 | 54.3 | 64.1 | 51.0 |
| (6) $c_n^r = JS$ | 85.9 | 54.3 | 64.2 | 51.1 |
| (7) $c_n^r = Euc$ | 86.0 | 54.4 | 64.1 | 51.2 |
| (8) w/o *distance* | 86.9 | 55.3 | 65.0 | 51.3 |
| (9) w/o $\hat{p}_r$ and $\hat{p}_n$ | 84.0 | 53.2 | 62.6 | 49.4 |
| CVC-MV (ours) | 87.2 | 55.7 | 65.3 | 51.3 |

Table 5: The ablation study on SQuAD (BERT-base). (1)-(4) are ablative settings for multi-task training (using CVC-IV); (5)-(9) are ablative settings related to CVC-MV.

observation indicates that this setting without all the shortcut branches cannot make a good trade-off on different adversarial attacks. The ablation study for MCQA is shown in Appendix.

**Extension to Natural Language Inference.** Our CVC method can also work on other NLP tasks like Natural Language Inference (NLI) task. Following the setting in previous works Clark et al. (2019), we train the model on MNLI Williams et al. (2018) and evaluate it on an adversarial set, HANS McCoy et al. (2019). We use the overlapped tokens in the hypothesis and premise as the only bias branch in the implementation of CVC. From the results shown in Table 12, we observe that CVC-MV outperforms CT by over 9% on the adversarial set, and achieves comparable performance compared to state-of-the-art methods.

## 6 Conclusions

We inspect the problem of fragility in QA models, and leverage the structural causal model to show that the crux is from shortcut correlations. To train robust QA models, we propose a novel training approach called Counterfactual Variable Control (CVC) and realize it based on a multi-task training pipeline. We conduct extensive experiments on multiple QA benchmarks, and show that CVC can achieve high robustness while being easy to interpret. Our future work is to enhance the structural causal model by considering subjective factors, *e.g.*, the preference of dataset annotators and the source of passages. These factors could be the confounders which may have the effects on the input variables and answer prediction simultaneously. For example, the tendency of the annotators or the crowdsourced workers. Such confounders may guide the model to conduct bias inference. Some intervention techniques can be applied to remove the effect of confounders.

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

## A  Algorithm

In Algorithm 1, we summarize the overall process of the proposed Counterfactual Variable Control (CVC) approach.

---

**Algorithm 1** Counterfactual Variable Control (CVC) algorithm

---

**Stage One: Multi-task Training**

**Input:** complete train set data $\mathcal{X}$ and $N$ different subsets of train set data $\{\mathcal{X}_n\}_{n=1}^N$

**Output:** $F^r$ with parameters $\theta^r$ and $\{F_n^s\}_{n=1}^N$ with parameters $\{\theta_n^s\}_{n=1}^N$

1: **for** batch in $\mathcal{X}$ and $\{\mathcal{X}_n\}_{n=1}^N$ **do**
2:    **for** $n$ in $\{1, ..., N\}$ **do**
3:       optimize $\theta_n^s$ with batch of $\mathcal{X}_n$ by Eq. equation 10;
4:    **end for**
5:    optimize $\theta^r$ with batch of $\mathcal{X}$ by Eq. equation 11 for MCQA (by $\mathcal{L}^{e2}$ in Eq. equation 12 for EQA);
6: **end for**

**Stage Two: Counterfactual Inference**

**Input:** $F^r$ with parameters $\theta^r$, $\{F_n^s\}_{n=1}^N$ with parameters $\{\theta_n^s\}_{n=1}^N$, complete target test data $\mathcal{X}'$ along with its subsets $\{\mathcal{X}_n'\}_{n=1}^N$ and a boolean $USE\_IV$.

**Output:** CVC inference result ($\{F_n^c\}_{n=1}^N$ with parameters $\{\theta_n^c\}_{n=1}^N$)

1: **if** $USE\_IV$ **then**
2:    compute CVC-IV inference result with target data by Eq. equation 14;
3: **else**
4:    optimize $\{\theta_n^c\}_{n=1}^N$ with $\mathcal{X}$ and $\{\mathcal{X}_n\}_{n=1}^N$ by Eq. equation 15, Eq. equation 16 and cross-entropy loss for QA task;
5:    compute CVC-MV inference result with target data $\mathcal{X}'$ and $\{\mathcal{X}_n'\}_{n=1}^N$ by Eq. equation 15 and Eq. equation 16;
6: **end if**

---

## B  MCQA-Specific Implementation

|                | MCTest | DREAM | RACE |
|----------------|--------|-------|------|
| Random guess   | 25.0   | 33.3  | 25.0 |
| Complete input | 68.9   | 61.5  | 64.7 |
| No $P$         | 24.2   | 32.8  | 41.6 |
| No $Q$         | 52.5   | 57.1  | 51.0 |
| No $P$, $Q$    | 22.4   | 33.4  | 34.7 |

Table 6: Accuracies (%) of conventional training BERT-base MCQA models tested with complete input. "No $X$" means the value of input variable $X$ is muted.

MCQA has two shortcut correlations (see Figure 2), *i.e.*, $Q \to A$ and $P \to A$[2]. We present the muting experiment results of MCQA in Table 6 that can reflect the strength of corresponding direct cause-effects. For example, the results on the row of "No $Q$" represent the performance of only using $P \to A$ and $O \to A$ shown in Figure 2 (b). We inspect them and notice that the effect from the former one is trivial and negligible compared to the latter. One may argue that $Q$ is an important cue to predict the answer. Actually, annotators intentively avoid any easy question-answer pairs when building MCQA datasets. For example, they include a person name in all options of questions about *who*. We thus assume $Q \to A$ has been eliminated during well-designed data collection and utilize one shortcut branch (*i.e.*, muting $Q$). Therefore, Eq. equation 11 and equation 12 are equivalent for MCQA ($N = 1$ and $w_n = 1$). Other MCQA-specific implementation details are the same with the official code of Devlin et al. (2019).

---

[2]$O \to A$ is not discussed here as $O$ is mandatory and can not be muted.

|  | SQuAD |
|---|---|
| Complete input | 88.1 |
| No $E$ | 59.4 |
| No $V$ | 55.1 |
| No $E$, $V$ | 15.3 |
| No $Q$ | 12.4 |

Table 7: F1 scores (%) of conventional training BERT-base EQA models tested with complete input. "No $X$" means the value of input variable $X$ is muted.

## C  EQA-Specific Implementation

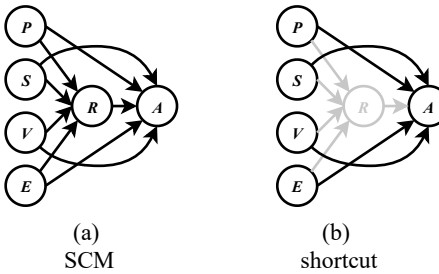

(a) SCM

(b) shortcut

Figure 4: The SCM for EQA task where $Q$ is decomposed to $S$, $V$ and $E$.

Different from MCQA, we propose to manually separate the *question* ($Q$) of EQA into corresponding parts: entities & nouns (*E*); verbs & adverbs (*V*); and the remaining stop words & punctuation marks (*S*). As shown in Figure 4, the SCM of EQA contains four input variables as $P$ (*passage*), $E$, $V$ and $S$. The comprehensive *reasoning* variable $R$ mediates between these four variables and *answer* $A$. The reason why we conduct this partition is twofold: (1) $P$ is mandatory for EQA. The lack of $P$ will result in an invalid prediction. To study the effects of $Q{\to}A$, what we can do is to split the variable $Q$ into partitions. (2) Our resulting $Q$ partitions are intuitive. $E$ and $V$ contain the most important semantic meanings. We inspect the empirical effects of all shortcut paths as shown in Table 7, and build shortcut branches with $N{=}2$ to represent all shortcut paths in Figure 4(b). The first shortcut branch takes $\mathcal{X}_1 = \{P, S, V\}$ as input and aims to learn $P, S, V{\to}A$. The second shortcut branch takes $\mathcal{X}_2 = \{P, S, E\}$ as input and learns $P, S, E{\to}A$. We empirically use $\mathcal{L}^{e2}$ to train EQA models. Other EQA-specific implementation details are the same with the official code of Devlin et al. (2019).

## D  Ablations on MCQA

Table 8 shows the MCQA results in 10 ablative settings. Specifically, we (1) use $\mathcal{X}_1 = \{Q, O\}$ as the input of the only shortcut branch; (2) use two shortcut branches, where the first one takes $\mathcal{X}_1 = \{P, O\}$ as input and the second one takes $\mathcal{X}_2 = \{Q, O\}$ as input, and deploy the $\mathcal{L}^e$ in Eq. equation 11; (3) use the same two shortcut branches as (2), but deploy the $\mathcal{L}^{e1}$ in Eq. equation 12; (4) use the same two shortcut branches as (2), but $\mathcal{L}^{e2}$ in Eq. equation 12 is used; The ablative setting of (5)-(9) on MCQA are the same as those used for EQA.

Results on (1)-(4) show that considering the shortcut branch with input $\{Q,O\}$ is not effective for the robustness of model. The reason is that this shortcut branch is hard to train, *i.e.*, not easy to converge (please refer to "MCQA-specific" and Table 6). Our empirical conclusions are as follows. Firstly, the shortcut branch with negligible effect magnitude can be ignored when designing the multi-branch architecture. Secondly, if no prior knowledge of the effect magnitude on each shortcut path (of SCM), using $\mathcal{L}^{e2}$ is the best choice. Results on (5)-(9) show the efficiency of our proposed $c$-adaptor.

| Ablative Setting | Test | Adv1 | Adv2 | Adv3 | Adv4 |
|---|---|---|---|---|---|
| (1) one modified Shct.br. | 68.3 | 63.1 | 58.0 | 24.8 | 56.5 |
| (2) two Shct.br. with $\mathcal{L}^e$ | 70.1 | 66.8 | 61.0 | 24.6 | 57.1 |
| (3) two Shct.br. with $\mathcal{L}^{e1}$ | 70.2 | 66.7 | 62.1 | 25.6 | 56.5 |
| (4) two Shct.br. with $\mathcal{L}^{e2}$ | 70.8 | 66.6 | 61.8 | 27.1 | 62.2 |
| CVC-IV (ours) | 69.4 | 70.0 | 65.4 | 28.7 | 59.9 |
| (5) same $c_n^r$ | 68.1 | 69.3 | 64.4 | 25.6 | 59.3 |
| (6) $c_n^r = JS$ | 70.1 | 67.0 | 61.9 | 20.8 | 62.2 |
| (7) $c_n^r = Euc$ | 69.8 | 67.7 | 61.9 | 22.3 | 60.5 |
| (8) w/o $distance$ | 66.1 | 67.9 | 65.2 | 27.8 | 61.0 |
| (9) w/o $\hat{p}_r$ and $\hat{p}_n$ | 65.6 | 66.3 | 64.8 | 27.4 | 59.9 |
| CVC-MV (ours) | 68.1 | 69.1 | 65.6 | 26.8 | 61.0 |

Table 8: The ablation study on MCTest (BERT-base). (1)-(4) are ablative settings for multi-task training (using CVC-IV inference). "Average" means the average performance on `Adv*` test sets; (5)-(9) are ablative settings related to CVC-MV inference.

| | | Test | Adv1 | Adv2 | Adv3 | Adv4 | A.G. |
|---|---|---|---|---|---|---|---|
| | CT | 71.0 | 70.6 | 72.1 | 42.5 | 60.5 | - |
| **Adv1** | CVC-IV | 71.7 | 73.3 | 74.9 | 49.2 | 63.8 | +3.9 |
| | CVC-MV | 71.6 | 72.9 | 74.8 | 48.0 | 62.7 | +3.2 |
| | CT | 72.3 | 73.0 | 75.1 | 50.1 | 63.3 | - |
| **Adv2** | CVC-IV | 71.8 | 73.8 | 76.2 | 59.8 | 65.5 | +3.5 |
| | CVC-MV | 71.8 | 74.2 | 76.6 | 61.1 | 65.5 | +3.9 |
| | CT | 67.5 | 62.7 | 59.9 | 70.9 | 57.1 | - |
| **Adv3** | CVC-IV | 67.6 | 64.5 | 62.4 | 70.2 | 61.6 | +2.0 |
| | CVC-MV | 66.8 | 63.7 | 62.3 | 70.3 | 60.5 | +1.5 |
| | CT | 69.8 | 65.4 | 60.2 | 27.7 | 63.3 | - |
| **Adv4** | CVC-IV | 69.9 | 66.2 | 62.4 | 32.7 | 61.0 | +1.4 |
| | CVC-MV | 67.5 | 65.6 | 62.4 | 25.4 | 66.7 | +0.9 |
| | CT | 70.5 | 72.1 | 74.1 | 72.5 | 63.4 | - |
| **All** | CVC-IV | 72.7 | 73.5 | 76.4 | 71.9 | 68.4 | +2.0 |
| | CVC-MV | **73.1** | **74.6** | **76.6** | **73.3** | **73.5** | **+4.0** |

Table 9: Accuracies (%) on the MCTest dataset, using different kinds of data augmentation in training with BERT-base. The leftmost column shows which type of adversarial attack for MCQA is used as data enhancement.

## E  Data Augmentation

Data augmentation with adversarial examples is an intuitive method to improve the model robustness Ribeiro et al. (2018); Jia & Liang (2017). We conduct experiments on the MCTest dataset to show the effect of augmentation adversarial data on CT, CVC-IV, and CVC-MV. Specifically, we augment the training data by generating adversarial samples following our adversarial attacks `Adv`. The results are shown in Table 9. Comparing Table 9 to the results without data augmentation (Table 1), we can observe that models get consistently improved via data augmentation. Comparing the results between CT and CVC, we find that CVC achieves further performance boosts for augmented models. For example, CVC-MV gains an average accuracy increase of 4.0% to "Add All" models when the training data are augmented with all the four kinds of adversarial examples. Note that it is high-cost and time consuming to conduct the data augmentation

experiments for EQA, because the adversarial attacks for EQA require a lot of human annotations and proofreading.

## F Efficiency

| | Training | | | | Inference | | | |
| | MCQA | | EQA | | MCQA | | EQA | |
| | Time | Parameter | Time | Parameter | Time | Parameter | Time | Parameter |
|---|---|---|---|---|---|---|---|---|
| CT | 1× | 1× | 1× | 1× | 1× | 1× | 1× | 1× |
| CVC-IV | 2× | 1.17× | 3× | 1.33× | 1× | 1× | 1× | 1× |
| CVC-MV | 2× | 1.17× | 3× | 1.33× | 2× | 1.17× | 3× | 1.33× |

Table 10: Efficiency comparison with convention training (CT) with respect to time and the number of parameters.

We show the summarized efficiency compared to conventional training (CT) in Table 10. While CVC-MV requires multiple runs on BERT (or RoBERTa) and consumes more computation during the inference stage, it has advantages over CVC-IV. Firstly, CVC-MV offers more interpretability for the robust QA model as demonstrated in the case studies (Figure 5 and Figure 6). CVC-MV illustrates the debiasing process based on causal inference by subtraction (CP-NP) while the result is directly given by CP for CVC-MV. Secondly, CVC-MV outperforms CVC-IV in certain cases, such as in the data augmentation scenario (Table 9) where all types of adversarial data are augmented during training. In addition, there are several ways to further reduce the computation cost of CVC-MV. Firstly, we can train the bias branch with a shallower model, such as the bottom 4 layers of BERT. Secondly, we can employ knowledge distillation to compress the bias branch even further.

## G Adversarial Dataset

| | Adv1 | Adv2 | Adv3 | Adv4 |
|---|---|---|---|---|
| MCTest | 840 | 840 | 835 | 177 |
| DREAM | 1119 | 911 | 2003 | 577 |
| RACE | 4892 | 4576 | 4876 | 1330 |
| SQuAD | 3560 | 1787 | 542 | - |

Table 11: Number of samples for the adversarial sets on four benchmarks.

We show the statistics for all the adversarial datasets in Table G. Building adversarial datasets for MCQA is a more straightforward process than for EQA. We devise several methods to add distractor options to the MCQA datasets, using options that have obvious word overlaps with the passage to confuse the model. The generation process of `AddVerb` for EQA is similar to that of `AddSent` Jia & Liang (2017). The differences consist of (i) `AddVerb` is used to evaluate the robustness of the model against verb attacks and (ii) `AddVerb` instances are annotated by a human expert linguist completely (raw version of `AddSent` is firstly generated by machine). Given a question-answer pair, the linguist creates a distracting `AddVerb` sentence in three steps:

- Replace the verb in the question with an antonym of this verb or an irrelevant verb. The verb is selected by the annotator to have an exact meaning.

- Create a fake answer with the same type as the ground-truth answer.

- Combine modified question and fake answer, and convert them into statement.

An illustration of the whole process is shown in Figure 5.

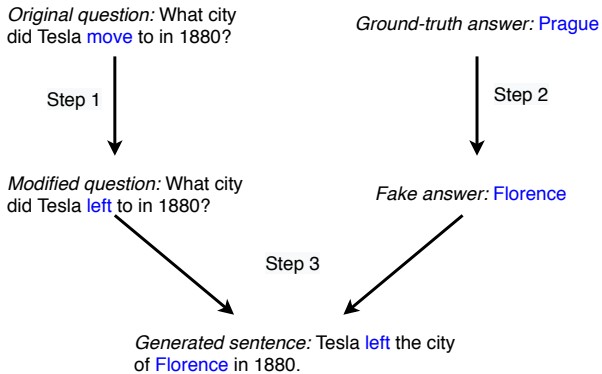

Figure 5: An illustration of the `AddVerb`. We use the instanced question-answer pair in Jia & Liang (2017) as an example.

# H   Case Studies

We show two examples as case studies to show the interpretability of our approach from two aspects: (1) the disentanglement of robust paths and shortcut in multi-branch architecture, (2) human-like counterfactual inference. Figure 6 and Figure 7 (in Appendix) illustrate two samples from MCQA and EQA respectively to demonstrate the underlying mechanism of CVC-IV and CVC-MV inference. In Figure 6, the conventional training method CT Devlin et al. (2019) merely aligns the words between *passage* and *options*. This action leads to the wrong choice $C$, which is a confusing choice generated by `Adv1`. In contrast, both CVC-IV and CVC-MV pick the right answer $D$. On the bottom blocks, we demonstrate the calculation on prediction logits during CVC-IV (Eq. 5) and CVC-MV (Eq. 6), respectively. We take the CVC-MV as an example to interpret this calculation. Both Normal Prediction (NP) $A_{p,q,o,r}$ and Counterfactual Prediction (CP) $A_{p,q,o,r^*}$ contain the logits of $A$, $B$, $C$ and $D$. The logit value of $C$ is from the word alignment shortcut and it is high in both NP and CP. It thus can be counteracted after the subtraction in CVC-MV. In contrast, the logit value of $D$ is from the comprehensive *reasoning*. When muting the corresponding variable $R$ (denoted by $r^*$ in CP $A_{p,q,o,r^*}$), this value must be reduced. Then it becomes evident after the subtraction in CVC-MV. The sample in Figure 7 on EQA can be interpreted in the same way. The only differences is that the "options" for EQA are tokens, *e.g.*, which token is the start position for answer span). Note that we normalize the bar chart (the result of the subtraction) for a clear visualization.

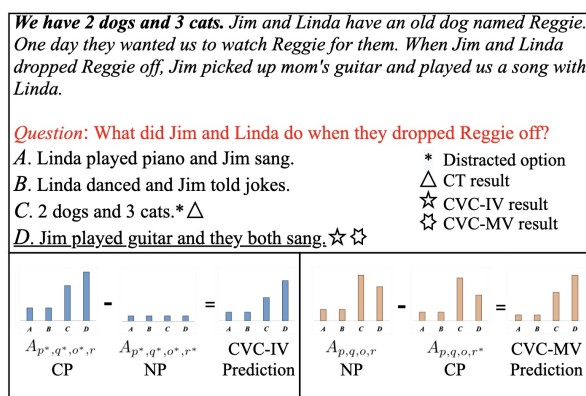

Figure 6: A case study of CVC on MCTest trained on official data. The ground truth is underlined.

*On the other hand, Luther also points out that the Ten Commandments when considered not as God's condemning judgment but as an expression of his eternal will, that is, of the natural law also positively teach how the Christian ought to live. This has traditionally been called the "third use of the law." For Luther, also Christ's life, when understood as an example,* **is nothing more than an illustration of the Ten Commandments**, *which a Christian should follow in his or her vocations on a daily basis.* **Luther denied Christ's life a dark story.**

*Question*: What did Luther consider Christ's life?

**Ground-truth answer:** *illustration of the Ten Commandments*
**CT result:** *a dark story*
**CVC-IV result:** *an illustration of the Ten Commandments,*
**CVC-MV result:** *an illustration of the Ten Commandments,*

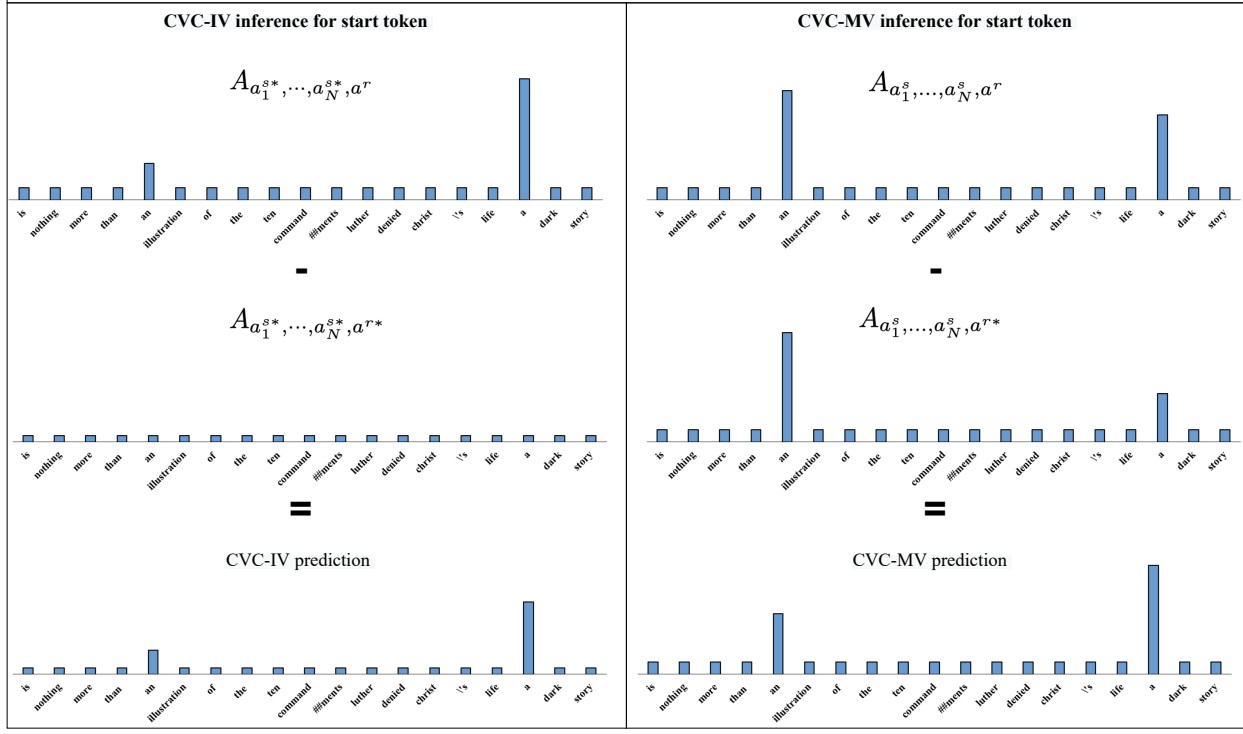

Figure 7: A case study of CVC on SQuAD trained on official data. The distracting sentence from `AddVerb` is underlined. Only bold tokens in passage are shown in bar chart due to limited page size.

|  | Matched Dev | HANS |
|---|---|---|
| CT | 84.2 | 62.4 |
| Reweight Clark et al. (2019) | 83.5 | 69.2 |
| Bias Product Clark et al. (2019) | 83.0 | 67.9 |
| Learned-Mixin Clark et al. (2019) | 84.3 | 64.0 |
| Learned-Mixin+H Clark et al. (2019) | 84.0 | 66.2 |
| DRiFt-HYPO He et al. (2019) | 84.3 | 67.1 |
| DRiFt-HAND He et al. (2019) | 81.7 | 68.7 |
| DRiFt-CBOW He et al. (2019) | 82.1 | 65.4 |
| Self-debias+Conf-reg Utama et al. (2020b) | 84.5 | 69.1 |
| Self-debias+Reweight Utama et al. (2020b) | 82.3 | 69.7 |
| Mind the Trade-off Utama et al. (2020a) | 84.3 | 70.3 |
| Forgettable$_{HANS}$ Yaghoobzadeh et al. (2019) | 84.3 | 70.4 |
| Forgettable$_{BoW}$ Yaghoobzadeh et al. (2019) | 83.4 | 71.2 |
| Forgettable$_{BiLSTM}$ Yaghoobzadeh et al. (2019) | 83.3 | 71.3 |
| End2End Ghaddar et al. (2021) | 83.2 | 71.2 |
| EIIL Yu et al. (2022) | 83.9 | 69.9 |
| CVC-IV | 82.9 | 70.0 |
| CVC-MV | 83.0 | **71.5** |

Table 12:  NLI accuracies (%) on Matched Dev and HANS. Our CVC methods are trained only on the original training data (MNLI) with BERT-base.

