# OpenReview forum: "Counterfactual Variable Control for Robust and Interpretable Question Answering"
_TMLR — Rejected by TMLR_

### Review · Reviewer_fZno · 2023-02-11

**Summary Of Contributions:**

The paper studies question answering through the lens of counterfactual variable control — investigating inference shortcuts of QA models and how that impact their robustness. They introduce two CVC inferences — one controlling the input variable and another controlling the mediator variable and evaluate their method on a suite of datasets, showing promising results (gains on out-of-domain, adversarily edited dataset with little harm to in-domain dataset). The motivation is good, and the overall paper is well-written.

**Audience:**

Yes

**Claims And Evidence:**

Yes

**Requested Changes:**

* Discussion of efficiency
* Clarification of comparison with related work

**Strengths And Weaknesses:**

Strengths:
* I am not aware of causal inference study on QA literature. While I find the concept question "causes" the answer a bit of a stretch, still it is a pretty interesting setting.
* Experimental results are positive.

Weaknesses:
* Discussion of efficiency is necessary. From what I understand, the proposed model runs inference multiple times on the same example, with different branches each processing different sets of inputs. How does this impact model training as well as inference time? I also wonder whether it provides some form of ensemble effects, compared to the baseline model.  Implementation detail mentions sharing first 10 layers for parameter efficiency for different branches, but further quantifying the model size / training & inference size would be helpful. While efficiency is not the focus of the work, as scaling often brings gains effortlessly these days, discussing efficiency will be good.
* The difference between existing work, such as Clark et al, should be further clarified. It mentions it as a “special case of CVC” in section 1. It mentions key difference as “systematic and explainable causal formulation”, but this is vague to me. How does Clark et al 2019 would compare with a single shortcut branch that considers simpler model instead of partial input model? If I understand correctly, Utama et al is quite similar to Clark et al — you should discuss these work more clearly when discussing results.
* Some parts of the paper are not very clear. I’m a bit confused about equation 14 — what does c^r_n and c^s_n mean and where do they come from? I am really confused there.. I conceptually understand the method, but not sure I understand its implementation fully. What is “r=R(p,q,o)” in Figure 3, for example?
* The process of the adversarial dataset should be better motivated and explained. How many examples in each dataset? How was verb selected? How was expert linguists recruited? Would these resources be released?

== minor comments and suggestions:
* I think the paper is a bit confusing in mentioning it as “inference” method, as it actually involves “training” multiple models — it’s not mere inference method which only changes inference procedure of already trained model. I think clarification will be helpful.
* I don’t think Figure 1 is very informative — this is not a new finding/contribution of this paper. Partial input settings have been studied in many prior work, and is fairly well known model can do this…  https://arxiv.org/pdf/1808.04926.pdf already showed this.
* SEQA/MCQA — The paper does not define what SEQA is? Is it supposed to mean span extraction question answering? Extractive QA might be better-known term.
* I was a bit bothered by the “Case Studies” section — this is impossible to make sense without referring to figures in the appendix. Also, I’m not sure looking two examples (probably cherry-picked) is a very useful analysis.
* Is QAInfomax (from 2019) really state-of-the-art? Might worth a bit more literature search?
* Discussing the generality/applicability of the method in other NLP tasks would be helpful. From what I see, method itself is fairly general and can be used for many kinds of classification task. How does this compare to counterfactual method introduced for NLI tasks / etc discussed in the related work?  -- Oh actually this is on Table 10 in the appendix! I think it should be referred to in the main text.

---

> ### Author Response · Authors · 2023-03-01
> **Response to Reviewer fZno**
>
> Thank you for your valuable feedback and comments. We have revised the paper in response to your suggestions, with the major changes or new contents highlighted in blue. Please note that the Span-extraction Question Answering (SEQA) has been modified to Extractive Question Answering (EQA) in the new version.
>
> Weakness 1-1 and Requested Change 1: Discussion of efficiency is necessary. From what I understand, the proposed model runs inference multiple times on the same example, with different branches each processing different sets of inputs. How does this impact model training as well as inference time? Implementation detail mentions sharing first 10 layers for parameter efficiency for different branches, but further quantifying the model size / training & inference size would be helpful. While efficiency is not the focus of the work, as scaling often brings gains effortlessly these days, discussing efficiency will be good.
>
> Response:
> We have included an additional section in Appendix F, along with Table 10, to further discuss the efficiency of CVC. The content of Appendix F and Table 10 is as follows:
>
> During the training stage, the computational cost of CVC is several times (1+number of shortcut branches) higher than Conventional Training (CT), increasing with the number of shortcut branches. However, it has the same computation cost compared to other baselines in Table 3 (except for CT) as they also train additional bias branch(es). During the inference stage, CVC-IV has the same computational cost as other baselines, while CVC-MV's computation cost is several times (1+number of shortcut branches) higher than others.
>
> From the perspective of parameters, the number of parameters for CVC-IV is the same as other baselines. However, the number slightly increases for CVC-MV (1/6 more parameters for one more branch), as most modules are shared (see Section 5.2 Implementation Details).
>
> While CVC-MV requires multiple runs on BERT (or RoBERTa) and consumes more computation during the inference stage, it has advantages over CVC-IV. Firstly, CVC-MV offers more interpretability for the robust QA model as demonstrated in the case studies (Figure 6 and Figure 7). CVC-MV illustrates the debiasing inference process based on causal inference by subtraction (CP-NP) while the result is directly given by CP for CVC-MV. Secondly, CVC-MV outperforms CVC-IV in certain cases, such as in the data augmentation scenario (Table 9) where all types of adversarial data are augmented during training.
>
> In addition, there are several ways to further reduce the computation cost of CVC-MV. Firstly, we can train the bias branch with a shallower model, such as the bottom 4 layers of BERT. Secondly, we can employ knowledge distillation to compress the bias branch even further.
>
> Weakness 1-2: I also wonder whether it provides some form of ensemble effects, compared to the baseline model.
>
> Response: CVC may appear similar to ensemble learning, but it is not considered a form of ensemble learning. CVC leverages results from multiple branches with different inputs based on the causal theory, such as the bias branch with incomplete inputs. In contrast, ensemble learning typically involves combining multiple models which have the same input data for an ensembled result. To further explore this, we conducted an experiment by training two MCQA models on the MCTest dataset and then ensembling them to vote for the final answer. The simple ensemble achieved an average gain (A.G.) of 0.4 points on adversarial sets, while CVC-IV and CVC-MV achieved an A.G. of over 6 points (Table 1).

---

> ### Author Response · Authors · 2023-03-01
> **Response to Reviewer fZno**
>
> Weakness 2 and Requested Change 2: (a) The difference between existing work, such as Clark et al, should be further clarified. It mentions it as a “special case of CVC” in section 1. (b) It mentions key difference as “systematic and explainable causal formulation”, but this is vague to me. (c) How does Clark et al 2019 would compare with a single shortcut branch that considers simpler model instead of partial input model? (d) If I understand correctly, Utama et al is quite similar to Clark et al — you should discuss these work more clearly when discussing results.
>
> Response to (a):
> We have included more explanation in section 1: “These ensemble-based debiasing methods also involve multi-branch training, with robust and bias branches, and only the robust branch during inference. It is similar in CVC-IV.”
>
> Response to (b):
> We have included more explanation in section 2: “Our first step was to formulate the QA task from a causal perspective using the Structural Causal Model (SCM) in order to understand the reasons behind the vulnerability of deep models. Building upon this analysis, we then proposed CVC to develop robust and interpretable QA models. CVC leverages the insights gained from the SCM analysis and offers an intuitive inference process for human interpretation.”
>
> Response to (c):
> On SQuAD dataset, Clark et al compare the method with a simpler model (TF-IDF model). The reason is that designing the partial input model is non-trivial for some NLP tasks. In our paper, we design two partial input bias branches for SEQA task with fruitful analysis and pre-experiments detailed in Appendix C.
>
> Response to (d):
> Yes, Utama et al is similar to Clark et al since they are ensemble-based debiasing methods.
> We have included more discuss in Comparison with Baselines and State-of-the-Arts part in section 5.3: “The main differences between these methods are the equations of ensembling in the training and the design of the bias model. For example, Clark et al adopts a TF-IDF model as the bias model on EQA while Utama et al employs an early-stopping model as the bias model.”
>
> Weakness 3: (a) Some parts of the paper are not very clear. I’m a bit confused about equation 14 — what does c^r_n and c^s_n mean and where do they come from? I am really confused there.. I conceptually understand the method, but not sure I understand its implementation fully. (b) What is “r=R(p,q,o)” in Figure 3, for example?
>
> Response: (a) Each element in $c^r_n$ or $c^s_n$ are the same constant derived from Eq. 9. Specifically, $c^r_n$ denotes the value of $\hat{\mathtt{p}}^r$ when the corresponding robust branch is fed null input, and thus the same constant is used to denote the logits in $c^r_n$. Similarly, $c^s_n$ corresponds to the value of $\hat{\mathtt{p}}_{n}^s$ when the corresponding shortcut branch is fed null input. We have included this in the revised version.
> (b) r=R(p,q,o) denotes the value of the comprehensive reasoning variable when all the inputs are not null values.
>
> Weakness 4: The process of the adversarial dataset should be better motivated and explained. How many examples in each dataset? How was verb selected? How was expert linguists recruited? Would these resources be released?
>
> Response: We show the statistics for all the adversarial datasets in Table 11 (a newly added table in the revised version). Building adversarial datasets for MCQA is a more straightforward process than for SEQA. We devise several methods to add distractor options to the MCQA datasets, using options that have obvious word overlaps with the passage to confuse the model. The generation process of AddVerb for SEQA is similar to that of AddSent [1]. The differences consist of (i) AddVerb is used to evaluate the robustness of the model against verb attacks and (ii) AddVerb instances are annotated by a human expert linguist completely (raw version of AddSent is firstly generated by machine). Given a question-answer pair, the linguist creates a distracting AddVerb sentence in three steps: First, replace the verb with in the question with an antonym of this verb or an irrelevant verb. The verb is selected by the annotator to have an exact meaning. Second, create a fake answer with the same type as the ground-truth answer. Finally, combine modified question and fake answer, and convert them into statement. An illustration of the whole process is shown in Figure 5 (the newly added figure in the revised version).
>
> The above content is included in the revised Appendix G.
>
> We recruit expert linguists from master's students majoring in linguistics. The cost for each sample is 1.2 US dollars. All the resources and code will be open-sourced upon acceptance.
>
> [1] Adversarial examples for evaluating reading comprehension systems, EMNLP 2017

---

> ### Author Response · Authors · 2023-03-01
> **Response to Reviewer fZno**
>
> Suggestion 1: I think the paper is a bit confusing in mentioning it as “inference” method, as it actually involves “training” multiple models — it’s not mere inference method which only changes inference procedure of already trained model. I think clarification will be helpful.
>
> Response: Thanks for the suggestion, we have revised the corresponding part in the abstract and the rest of the paper. For example, we revised “a novel inference method called Counterfactual Variable Control (CVC)” to “a novel approach called Counterfactual Variable Control (CVC)”.
>
> Suggestion 2: I don’t think Figure 1 is very informative — this is not a new finding/contribution of this paper. Partial input settings have been studied in many prior work, and is fairly well known model can do this… https://arxiv.org/pdf/1808.04926.pdf already showed this.
>
> Response: While we acknowledge that some prior works have conducted similar experiments, such as the study referenced in https://arxiv.org/pdf/1808.04926.pdf, we emphasize that only Figure 1 is not a contribution of our paper. Figure 1 serves as a motivation for our study. Our primary contribution is the systematic and explainable CVC method for debiasing QA models, derived from causal inference.
>
> Suggestion 3: SEQA/MCQA — The paper does not define what SEQA is? Is it supposed to mean span extraction question answering? Extractive QA might be better-known term.
>
> Response: Thanks for the suggestion, we have added a definition of extractive QA and replaced all SEQA in our paper with Extractive Question Answering (EQA).
>
> Suggestion 4: I was a bit bothered by the “Case Studies” section — this is impossible to make sense without referring to figures in the appendix. Also, I’m not sure looking two examples (probably cherry-picked) is a very useful analysis.
>
> Response:
> Sorry that we need to move the figures to the appendix due to the page limits. To facilitate the reading, we have moved the case study part to appendix H near the two figures. We would like to highlight that the two examples are to illustrate how the inference stage of CVC is conducted with interpretability, not quantitative analysis. For example, the distracted option may have a very high probability on both NP and CP while a low probability after the subtraction on CVC-MV in Figure 6.
>
> Suggestion 5:
> Is QAInfomax (from 2019) really state-of-the-art? Might worth a bit more literature search?
>
> Response:
> Thanks for the suggestion, we added results from methods published in 2021-2022 in Table 2 for EQA task. In addition, we also included more results from the methods published in 2021-2022 Table 12 for NLI task.
>
> Suggestion 6:
> Discussing the generality/applicability of the method in other NLP tasks would be helpful. From what I see, method itself is fairly general and can be used for many kinds of classification task. How does this compare to counterfactual method introduced for NLI tasks / etc discussed in the related work? -- Oh actually this is on Table 10 in the appendix! I think it should be referred to in the main text.
>
> Response:
> Thanks for the suggestion, we have moved the NLI part to the main text (the Table is remained in Appendix due to the page limits).

---

### Review · Reviewer_PEsL · 2023-02-11

**Summary Of Contributions:**

The paper aims to build a more robust multi-choice question answering model that is less affected by the dataset bias, e.g., the model being able to choose the right answer from the options even when the evidence passage is not provided. They propose “Counterfactual Variable Control” with two variants, which is largely inspired by the causality in QA. Intuitively, the first variant takes the counterfactual (muted) input and maximizes the logits assigned to the original (gold) answers over the bias-preferred answer, and the second variant takes the original input and maximizes the logits assigned to the original (gold) answers) over the bias-preferred answer. Evaluation was done on MCQA and SEQ with various adversarial attacks designed to evaluate the robustness of the models. Both variants of the proposed methods achieve significant gains with adversarial attacks, with 1.6%–7.6% improvement in absolute over the baseline, while retaining its accuracy on the original test set.

In summary, the contributions are:
* A new formulation of the multi-choice question answering task with respect to casualty.
* A new training method that improves robustness of the QA model using the intuition drawn from the causality.
* Evaluation on a range of backbone models and adversarial attacks, showing consistent improvements made by the proposed models.


**Audience:**

Yes

**Broader Impact Concerns:**

I do not think there is any concern on the ethical implementation of the work.

**Claims And Evidence:**

Yes

**Requested Changes:**

* I do not have a specific request in the experimental section, since I think they are already comprehensive and the results are very impressive.
* However, I think the paper would greatly improve with better writing of Section 1, 3 and 4 overall, as mentioned in the "weakness" section above -- in particular, to make a tighter connection between plain text intuition, causality inference, and actual implementation of the model.

**Strengths And Weaknesses:**

Strengths
* The paper made connections between the multi-choice QA and causality inference, which is underexplored in prior literature.
* The method proposed in the paper is simple and is intuitive, and is shown to be empirically strong.
* Evaluation is comprehensive, including multiple datasets, multiple backbone LMs, and various adversarial attacks. The paper also includes comparison with strong baselines from prior work.

Weakness
* The approach is strictly limited to multi-choice question answering.
* The connection between multi-choice QA and causality inference was not made in a comprehensive manner – it was only mentioned in the Introduction but not in other sections. Thus, it is not easy for readers who are unfamiliar with causality inference to understand the intuition. In particular, in Section 3, every component of the methodology is explained with the perspective of the causality inference, with no intuition provided in plain text, or no concrete definition on what each component means, e.g., comprehensive reasoning "R", which is the most critical component in the method.
* Moreover, only in Section 4 the paper starts to describe how the method is implemented, but again with loose connection with Section 3, because the description focuses on the overall architecture of the model rather than explicit definition of how "R" is implemented, etc.
* For these two reasons above, I have an impression that, while the paper tries to explain the idea with causality, there is a disconnection between their intuition using causality inference and how the model is actually implemented. It would have been much more helpful if intuition behind the method is tightly connected to each other -- from intuition in plain text, intuition based on causality inference, and the actual implementation.

---

> ### Author Response · Authors · 2023-03-01
> **Response to Reviewer PEsL**
>
> Thank you for your valuable feedback and comments. We have revised the paper in response to your suggestions, with the major changes or new contents highlighted in blue. Please note that the Span-extraction Question Answering (SEQA) has been modified to Extractive Question Answering (EQA) in the new version.
>
> Weakness 1: The approach is strictly limited to multi-choice question answering.
>
> Response: We also apply CVC method to Extraction Question Answering (EQA) tasks, i.e., SQuAD dataset, and Natural Language Inference (NLI) tasks, i.e., MNLI dataset. The implementation and main experimental results of CVC for EQA are presented in Table 2, Table 4, and Appendix C. CVC's implementation and main experimental results for NLI are provided in Section 5.3 - Extension to Natural Language Inference (moved to the main text in the new version, previously in the Appendix) and Table 12.
>
> Weakness 2:  The connection between multi-choice QA and causality inference was not made in a comprehensive manner – it was only mentioned in the Introduction but not in other sections. Thus, it is not easy for readers who are unfamiliar with causality inference to understand the intuition. In particular, in Section 3, every component of the methodology is explained with the perspective of the causality inference, with no intuition provided in plain text, or no concrete definition on what each component means, e.g., comprehensive reasoning "R", which is the most critical component in the method.
>
> Response:
> We would like to emphasize that we utilize multiple-choice question answering (MCQA) as a case study from the introduction (section 1) through to the implementation (section 4) of our proposed method. To provide further clarity, we have added additional guidance at the beginning of Section 3 and Section 4 in the revised version as follows:
>
> At the beginning of section 3: “In this section, we continually use multi-choice question answering (MCQA) (and its SCM in Figure 2 as a case study of QA tasks and introduce our proposed Counterfactual Variable Control (CVC) on the level of SCM. In the next section, we turn to the implementation that how to model SCM and conduct the CVC using the deep model.”
>
> At the beginning of section 4: “Building on Section 3, we continue to take MCQA and its corresponding SCM in Figure 2 as the example in this section.”
>
> We also included more explanation in section 3 to bridge the gap between causal inference and MCQA as well as explain the meaning of R:
>
> At the beginning of section 3.2: “As previously stated, our objective is to preserve only the robust prediction derived from comprehensive reasoning and exclude shortcut correlations. Consequently, the goal is to measure the effect of comprehensive reasoning R (Noted that R variable is a virtual variable compared to other variables and denotes robust and comprehensive reasoning). Motivated by the theory of causality, CVC can be realized by comparing the fact and its counterpart, i.e., estimating the difference between the normal prediction (NP) and the counterfactual prediction (CP). Intuitively, the importance of the effect of a variable on the resulting variable can be revealed by controlled experiments.
>
> Explanations for CVC-IV and CVC-MV after Eq. 5 and Eq. 6: “In CVC-IV equation, the first term is CP and the second term is NP. We measure the effect of R by comparing the two scenarios where the states of R are different. In CVC-MV, the first term is NP and the second term is CP. We measure the effect of R by comparing the two scenarios where the states of R are different.”
>
> Weakness 3: Moreover, only in Section 4 the paper starts to describe how the method is implemented, but again with loose connection with Section 3, because the description focuses on the overall architecture of the model rather than explicit definition of how "R" is implemented, etc.
>
> Response: We have added connection content as mentioned in response to weakness 2 in section 4. Part of the paths in SCM (Figure 2) including R, i.e., P,Q,O->R->A, is implemented using the robust branch, which is shown in the corresponding part of section 4.1. For clarity, we have augmented the explanation in this part as follows:
>
> In the P,Q,O->R->A part of section 4.1: “We explain that R can be regarded as the hidden state of the top layers before the classifier on the robust branch (see Figure 3). This illustrates the features of the text after being processed by the deep model's comprehensive reasoning ability.”

---

> > ### Author Response · Authors · 2023-03-01
> > **Response to Reviewer PEsL**
> >
> > Weakness 4: For these two reasons above, I have an impression that, while the paper tries to explain the idea with causality, there is a disconnection between their intuition using causality inference and how the model is actually implemented. It would have been much more helpful if intuition behind the method is tightly connected to each other -- from intuition in plain text, intuition based on causality inference, and the actual implementation.
> >
> > Response: Thanks for the suggestion! We have revised the paper as mentioned above.

---

### Review · Reviewer_MNHZ · 2023-02-18

**Summary Of Contributions:**

The paper proposed a method Counterfactual Variable Control (CVC) to improve models' robustness. They proposed two methods, CVC-IV and CVS-MV, which controls input variables and mediator variables respectively. Results show that both methods improves over baselines without adversarial examples and other adversarial training methods.

**Audience:**

Yes

**Broader Impact Concerns:**

I don't have any concerns on the ethical implications.

**Claims And Evidence:**

Yes

**Requested Changes:**

Overall the paper is clearly written and the claims in the paper are well supported. No major changes needed. Please consider replying to the questions and comments above.

**Strengths And Weaknesses:**

The paper proposed a method to reduce shortcut correlations when making predictions. They clearly showed that using the novel training mechanism with the multi-branch network architecture can effectively improve models performance over adversarial examples. The problem which the paper solved is an important topic in building NLP systems.

A few questions/comments:

1. For SEQA, do you need "options" during training, e.g. Eq. 14 and Eq. 15?
2. Also for SEQA, how do you mine the shortcut paths? Please provide more details.
3. How much is the additional computation cost for training with the CVC modules? From what I read, predicting each example requires multiple runs on BERT (or RoBERTa). How does it compare with other baselines in Table 3? How many shortcut branches are used for each example?

---

> ### Author Response · Authors · 2023-02-27
> **Response to Reiewer MNHZ**
>
> Thank you for your valuable feedback and comments.
>
> Q1: For SEQA, do you need "options" during training, e.g. Eq. 14 and Eq. 15?
>
> Answer: No, the option is not involved in SEQA. The inference process presented in Section 4 (Eq. 14 and Eq. 15) is an illustration of MCQA. Figure 4 (in the Appendix) displays the causal graph for SEQA, where the components for SEQA are shown.
>
> Q2: Also for SEQA, how do you mine the shortcut paths? Please provide more details.
>
> Answer: As shown in Figure 4, the SCM of SEQA contains four input variables P (passage), E, V, and S. The comprehensive reasoning variable R mediates between these four variables and answer A. The reason why we conduct this partition is twofold:
> (1) P is mandatory for SEQA. The lack of P will result in an invalid prediction. To study the effects of Q->A, what we can do is split the variable Q into partitions.
> (2) Our resulting Q partitions are intuitive. E and V contain the most important semantic meanings. We inspect the empirical effects of all shortcut paths as shown in Table 7.
> For more details, please kindly refer to Appendix C (we replaced SEQA with EQA following the suggestion from another Reviewer), Figure 4, and Table 7.
>
> Q3: How much is the additional computation cost for training with the CVC modules? From what I read, predicting each example requires multiple runs on BERT (or RoBERTa). How does it compare with other baselines in Table 3? How many shortcut branches are used for each example?
>
> Answer: During the training stage, the computational cost of CVC is several times (1+number of shortcut branches) higher than Conventional Training (CT), increasing with the number of shortcut branches. However, it has the same computation cost compared to other baselines in Table 3 (except for CT) as they also train additional bias branch(es). During the inference stage, CVC-IV has the same computational cost as other baselines, while CVC-MV's computation cost is several times (1+number of shortcut branches) higher than others.
>
> From the perspective of parameters, the number of parameters for CVC-IV is the same as other baselines. However, the number slightly increases for CVC-MV (1/6 more parameters for one more branch), as most modules are shared (see Section 5.2 Implementation Details).
>
> While CVC-MV requires multiple runs on BERT (or RoBERTa) and consumes more computation during the inference stage, it has advantages over CVC-IV. Firstly, CVC-MV offers more interpretability for the robust QA model as demonstrated in the case studies (Figure 6 and Figure 7). CVC-MV illustrates the debiasing inference process based on causal inference by subtraction (CP-NP) while the result is directly given by CP for CVC-MV. Secondly, CVC-MV outperforms CVC-IV in certain cases, such as in the data augmentation scenario (Table 9) where all types of adversarial data are augmented during training.
>
> We use one shortcut branch for MCQA and two for SEQA, respectively. Please kindly refer to Appendix B and C for more details.

---

### Decision · Action_Editors · 2023-04-20

**Recommendation:** Reject

**Comment:**

The paper attempts to improve question answering (QA) methods for machine reading comprehension. In this regards, the authors first identify the problem of “shortcut capability” of current QA models. As a solution to this problem, the paper propose to disentangle shortcut correlations and reasoning by introducing counterfactuals as a control. The empirical results in the paper seem to demonstrate that the proposed approach is robust against various adversarial attacks in QA tasks. We thank the authors and reviewers for actively engaging in the discussion and introduce new results towards making the paper better. However, the reviewers have concerns about: 1) positioning with existing literature, 2) efficiency vs gain in performance seems to be not favorable (does the conclusion hold as we scale to larger models because gains seem to keep diminishing but cost keeps increasing?), 3) writing and presentation of the paper, which needs improvement according to all reviewers. Novelty is not a requirement for TMLR, but accuracy and clarity are. Thus, unfortunately I cannot recommend an acceptance without another round of review of the paper, but I strongly encourage authors to resubmit after improving the writing and providing a more complete picture around motivation and actual model, the positioning with respect to prior works, and if the conclusion holds with scaling.

**Audience:**

Yes, NLP community, especially those focusing on Question-Answering should be interested in knowing the findings of this paper.

**Claims And Evidence:**

There might be some gap between the motivation/intuition and modeling/experiments carried out as pointed by reviewers e.g. "fundamental disconnectivity between the intuition from causality inference and the method proposed in this paper, which I don't think the new revision addresses", "confusing in mentioning it as inference method, as it actually involves training multiple models", etc.